# Putative Endoplasmic Reticulum Stress Inducers Enhance Triacylglycerol Accumulation in *Chlorella sorokiniana*

**DOI:** 10.3390/bioengineering12050452

**Published:** 2025-04-25

**Authors:** Yoomi Roh, Sujeong Je, Naeun Sheen, Chang Hun Shin, Yasuyo Yamaoka

**Affiliations:** 1Division of Biotechnology, The Catholic University of Korea, Bucheon 14662, Republic of Korea; yoomirohh@gmail.com (Y.R.); sujeongh.je@gmail.com (S.J.); 2Chong Kun Dang Bio (CKDBiO) Research Institute, Ansan 15604, Republic of Korea; naeun@ckdbio.com (N.S.); chshin@ckdbio.com (C.H.S.)

**Keywords:** *Chlorella*, ER stress, lipid, TAG

## Abstract

*Chlorella*, recognized for its high lipid and protein content, is increasingly studied for its potential in the food and bio industries. To enhance its production and understand the underlying mechanisms of lipid accumulation, this study investigated the role of endoplasmic reticulum (ER) stress in modulating lipid metabolism in *Chlorella sorokiniana* UTEX 2714, using six putative ER stress inducers: 2-deoxy-D-glucose (2-DG), dithiothreitol (DTT), tunicamycin (TM), thapsigargin (TG), brefeldin A (BFA), and monensin (Mon). The results showed that 2-DG, DTT, TM, BFA, and Mon significantly inhibited cell growth in *C. sorokiniana*. Treatment with 2-DG, DTT, TM, BFA, or Mon resulted in substantial increases in the triacylglycerol (TAG) to total fatty acid (tFA) ratio, with fold changes of 14.8, 7.9, 6.2, 10.1, and 8.9, respectively. Among the tFAs, cells treated with these compounds exhibited higher levels of saturated fatty acids and lower levels of polyunsaturated fatty acids (PUFAs). In contrast, the fatty acid composition of TAGs showed the opposite trend, with relative enrichment in PUFAs. This study enhances our understanding of *Chlorella* lipid metabolism, providing valuable insights for optimizing lipid production, particularly TAGs enriched with PUFA content, for applications in functional foods, nutraceuticals, and sustainable bioresources.

## 1. Introduction

Microalgae have been investigated as a potential sustainable resource for various industries due to their ability to produce bioactive compounds [1]. Among these, lipids are one of the most valuable products obtained from microalgae and can be used to produce biofuels. Triacylglycerol (TAG) is a key microalgal lipid, serving as its primary energy storage molecule and an essential feedstock for biofuel production [2]. Microalgae can accumulate TAG under specific conditions, such as nitrogen starvation or high light intensity, making them promising candidates for sustainable biofuel applications [3]. Enhancing TAG productivity through metabolic engineering or optimized cultivation strategies could enable scalable and efficient biofuel production, reducing reliance on fossil fuels while supporting carbon-neutral energy solutions. Many microalgae species are known for their rapid growth and capacity to produce a large amount of biomass. Additionally, under various stress conditions, such as nutrient starvation, they can accumulate lipids which can then be converted into biodiesels [4]. Among the many species of microalgae used for biotechnological applications, *Chlorella* has recently drawn considerable attention. Among the various stress conditions, nitrogen starvation is widely recognized as one of the most effective conditions to trigger TAG accumulation in microalgae. However, nitrogen starvation impairs photosynthesis, resulting in a reduction in biomass production. Moreover, in industrial environments, the process of nitrogen removal is energy-intensive, time-consuming, and costly [5]. To address this challenge, it is essential to develop alternative strategies for inducing lipid accumulation in microalgae without compromising biomass productivity or increasing operational costs.

The endoplasmic reticulum (ER) plays a crucial role in protein folding, lipid biosynthesis, and maintaining cellular homeostasis. Under normal conditions, the ER ensures that newly synthesized proteins are properly folded, assembled, and transported. However, various physiological or environmental stresses can disrupt such ER functions, leading to the accumulation of misfolded or unfolded proteins, a condition known as ER stress (as reviewed in [6,7]). To mitigate ER stress, cells activate a signaling pathway called the unfolded protein response (UPR), which reduces protein synthesis, enhances chaperone activity to aid in folding, and promotes the degradation of misfolded proteins. A previous study investigated the mechanism of seven different compounds that induce ER stress in animal cells: 2-deoxyglucose (2-DG), dithiothreitol (DTT), tunicamycin (TM), thapsigargin (TG), brefeldin A (BFA), monensin (Mon), and eeyarestatin I [8]. 2-DG is a synthetic glucose analog in which the C-2 hydroxyl group has been replaced with hydrogen. After entering the cell via glucose transporters, 2-DG is converted by hexokinase into phosphorylated 2-DG. This inhibits hexokinases non-competitively, reducing ATP and lactate production, ultimately inhibiting cell growth [9]. DTT inhibits protein disulfide bond formation, potentially activating the UPR. TM induces ER stress by inhibiting N-linked glycosylation, leading to the accumulation of unfolded proteins in ER. TG disrupts ER Ca^2+^ homeostasis by inhibiting sarco/endoplasmic reticulum Ca^2+^-ATPase (SERCA). Ca^2+^ homeostasis is important for the activation of ER chaperones [10]. BFA inhibits vesicle trafficking from the ER to the Golgi, resulting in the accumulation of unfolded proteins in ER and triggering the UPR. Mon is a Na^+^/H^+^ ionophore that disrupts the pH balance of the ER and the Golgi. The pH in the ER lumen is critical for protein folding and vesicle trafficking, and its disruption contributes to ER stress. Lastly, eeyarestatin I, an inhibitor of ERAD, has been reported to induce the UPR [11].

ER stress has been shown to initiate lipid droplet formation through several distinct, experimentally supported mechanisms across different organisms. In yeast, chemical stressors such as TM and BFA induce lipid droplet formation independently of the Ire1p pathway [12]. In mammalian systems, ER stress activates a transcriptional response involving SREBP-1c and other lipogenic regulators, thereby promoting TAG synthesis [13,14,15]. In microalgae, ER stress appears to affect not only protein homeostasis but also lipid metabolic pathways [16,17,18]. In *Chlamydomonas reinhardtii*, for example, TM-induced ER stress activates the IRE1/bZIP1 pathway, leading to increased levels of ER membrane-specific lipid species such as diacylglyceryltrimethylhomoserine (DGTS) and ER-enriched fatty acids like pinolenic acid (18:3Δ5,9,12) [16]. Our previous work further demonstrated that the ER stress inducers BFA and DTT specifically promote accumulation of storage lipids, particularly TAGs, accompanied by upregulation of acyltransferase genes such as *DGTT1* and *DGAT* [17].

In this study, we investigated the effects of six known ER stress–inducing compounds on the growth and lipid metabolism of *Chlorella sorokiniana* UTEX 2714, based on the hypothesis that ER stress promotes TAG accumulation through conserved stress response pathways. This strain was previously identified as *Chlorella sorokiniana* [19]. Treatment with 2-DG, DTT, TM, BFA, or Mon significantly inhibited cell growth. Lipid analysis revealed that 2-DG, DTT, TM, BFA, and Mon induced TAG accumulation in UTEX 2714 cells. Notably, we demonstrated for the first time that 2-DG efficiently induces TAG synthesis in *Chlorella*, a microalga capable of glucose uptake. These results reveal *Chlorella*-specific characteristics of lipid metabolism in response to putative ER stress compounds, thereby broadening our understanding of how ER stress influences lipid biosynthesis in microalgae.

## 2. Materials and Methods

### 2.1. Strain and Culture Conditions

*Chlorella sorokiniana* UTEX 2714 was obtained from the Culture Collection of Algae at the University of Texas (Austin, TX, USA). *Chlorella* cells were cultured in Tris-acetate-phosphate (TAP) medium at 23 °C under continuous light (75 μmol photons m^−2^ s^−1^) with constant shaking at 180 rpm. TAP medium consists of Tris base, NH_4_Cl, MgSO_4_·7H₂O, CaCl₂·2H_2_O, K_2_HPO_4_, and KH_2_PO_4_, and acetic acid. The pH was adjusted to 7.0 using HCl. Additionally, a trace elements solution was added, containing ZnSO_4_·7H_2_O, H₃BO₃, MnCl_2_·4H_2_O, CuSO_4_·5H_2_O, Na_2_MoO_4_·2H_2_O, CoCl_2_·6H_2_O, FeCl₃·6H_2_O, and Na_2_EDTA·2H_2_O [20]. Cells in the exponential growth phase were treated with the indicated concentrations of various ER stress compound, which were prepared as follows:2-DG: Diluted from a 2 M stock solution in water (Sigma-Aldrich, St. Louis, MO, USA).DTT: Diluted from a 1 M stock solution in water (Duchefa Biochemie, Haarlem, The Netherlands).TM: Diluted from a 5 mg/mL stock solution in 0.05 N NaOH (Sigma-Aldrich).TG: Diluted from a 5 mM stock solution in dimethyl sulfoxide (DMSO) (Sigma-Aldrich).BFA: Diluted from a 0.02 M stock solution in DMSO (Sigma-Aldrich).Mon: Diluted from a 0.03 M stock solution in DMSO (Sigma-Aldrich).

### 2.2. Spot Test for the Viability of Chlorella Cells

A spot test was performed with UTEX 2714 cells. Cells in the exponential growth phase were treated with 1, 5, and 10 mM of 2-DG; 1, 5, and 10 mM of DTT; 0.5, 1, and 5 μg/mL of TM; 0.1, 0.5, and 1 μM of TG; 1, 5, and 10 μM of BFA; and 10, 50, and 100 μM of Mon. After treatment, 5 μL of each treated cell suspension was spotted on TAP medium and then incubated for 48 h under standard growth conditions.

### 2.3. Measurement of Lipid Content and Observation of Lipid Droplets by Nile Red Staining

To quantify lipid droplets, cells were stained with 1 µg/mL Nile red (Sigma-Aldrich) diluted from a 0.1 mg/mL stock solution in acetone and incubated for 20 min in darkness at room temperature. The fluorescence intensity (FI) was measured using a microplate reader (Infinite M200 PRO; TECAN, Port Melbourne, VIC, Australia) with a 488 nm excitation filter and a 565 nm emission filter. The FI values were normalized to optical density at 750 nm (OD_750_). Confocal laser scanning microscopy was performed using an LSM 900 confocal microscope (Zeiss, Jena, Germany), and Nile red fluorescence was observed by excitation with a 488 nm laser and detection at 520/530 nm.

### 2.4. Lipid Extraction and Quantification

Lipids were extracted from the samples and analyzed as described previously [16]. Briefly, 10 mL of *Chlorella* cell culture was harvested by centrifugation at 2000× *g* for 5 min at room temperature and used for total lipid extraction. To quantify TAGs, 1 mg of the total lipid extract was separated on a TLC plate using the solvent mixture hexane:diethyl ether:acetic acid (80:30:1, *v*/*v*). Lipid spots were visualized under ultraviolet light after spraying with 0.01% (*w*/*v*) primuline (Sigma-Aldrich) dissolved in acetone:water (4:1, *v*/*v*). Lipid bands were recovered from the plate and quantified by gas chromatography with flame ionization detection (GC-FID) (GC-2030, Shimazu, Kyoto, Japan). The gas chromatograph was equipped with an HP-INNOWax capillary column (30 m × 0.25 mm, 0.25 μm film thickness, Agilent Technologies, Santa Clara, CA, USA) after transesterification. A capillary column was temperature-programmed from 185 °C to 230 °C at 10 °C min^–1^ with helium as the carrier gas. One microliter of sample was injected in a 270 °C inlet with a 20:1 split ratio. Methyl ester derivatives were detected using an FID at 270 °C.

## 3. Results

### 3.1. Compounds Reported to Induce ER Stress Affect the Growth of C. sorokiniana

To investigate the ER stress response in *C. sorokiniana* UTEX 2714, we analyzed the effects of six compounds reported to induce ER stress on its growth. The compounds tested, 2-deoxy-D-glucose (2-DG), dithiothreitol (DTT), tunicamycin (TM), thapsigargin (TG), brefeldin A (BFA), and monensin (Mon), are known to induce ER stress either by disrupting protein folding and causing accumulation of unfolded proteins in the ER lumen or by inhibiting vesicle transport associated with ER membranes through distinct mechanisms (Figure 1). After 48 h of treatment, cultures exposed to 2-DG, DTT, TM, BFA, and Mon exhibited visible chlorosis (Figure 2A). Chlorosis is a common response to stress in photosynthetic organisms and indicates impaired cellular function. However, no noticeable changes were detected in cells treated with TG, suggesting that this compound did not elicit a significant stress response under the conditions tested. To further quantify the growth inhibition, we measured the optical density (OD_750_) of *C. sorokiniana* cultures after 48 h of treatment with various concentrations of these compounds. The OD_750_ values significantly decreased following treatment with 2-DG, DTT, TM, BFA, and Mon (Figure 2B), consistent with the observed chlorosis (Figure 2A). Notably, TG treatment had no effect on OD_750_, further confirming its lack of impact on *C. sorokiniana* under these experimental conditions. To test the viability of *C. sorokiniana* cells, treated cells were spotted on the TAP agar plates without compounds and incubated for 48 h. A remarkable decrease in viability was observed for cells exposed to 2-DG, DTT, TM, BFA, and Mon, with the degree of viability reduction depending on the concentration of the respective compound (Figure 2C). This dose-dependent effect indicates that increasing concentrations of these compounds exacerbate their cytotoxic effects, likely due to heightened ER stress. In contrast, cells treated with TG showed no decrease in viability compared to untreated controls. These findings demonstrate that *C. sorokiniana* UTEX 2714 cells are sensitive to compounds previously reported to induce ER stress, including 2-DG, DTT, TM, BFA, and Mon, as evidenced by chlorosis, lower optical density, and reduced viability.

### 3.2. Lipid Droplet Accumulation Induced by 2-DG, DTT, BFA, TM, and Mon in C. sorokiniana cells

Previous studies have reported that ER stress–inducing compounds such as DTT, TM, and BFA promote the expression of transcripts related to lipid biosynthesis in *Chlamydomonas reinhardtii* [17]. Additionally, BFA treatment has been shown to induce lipid droplet formation in *Chlorella vulgaris* [21]. Based on these findings, we hypothesized that ER stress–inducing compounds that regulate the growth of *C. sorokiniana* would also trigger lipid droplet accumulation. To examine this hypothesis, we used Nile red, a lipid droplet–specific fluorescent dye, to examine lipid droplet accumulation in cells treated with ER stress–inducing compounds. After 48 h of treatment, cells exposed to various concentrations of 2-DG, DTT, TM, BFA, and Mon showed significant lipid droplet accumulation compared to the control cells (Figure 3). This result suggests a direct link among ER stress, cell growth inhibition, and lipid droplet formation in *C. sorokiniana* UTEX 2714.

### 3.3. Microscopy Confirms Lipid Droplet Formation After 48 h of Treatment with Putative ER Stress Inducers

For further analysis, we conducted microscopic observation of Nile red–stained cells to visualize lipid droplet formation. Concentrations of 2-DG, DTT, TM, BFA, and Mon were selected based on their ability to maintain cell viability comparable to that observed in Figure 2, enabling meaningful comparative analysis. We also included the highest tested concentration of TG (1 μM), which did not produce a noticeable effect under the conditions used (Figure 2). The concentrations applied were: 5 mM 2-DG, 10 mM DTT, 5 μg/mL TM, 1 μM TG, 5 μM BFA, and 50 μM Mon. Microscopic observation confirmed that the cells treated with 2-DG, DTT, TM, BFA, and Mon showed lipid droplet accumulation after 48 h (Figure 4). In contrast, TG treatment did not induce lipid droplet formation, aligning with the results of the Nile red fluorescence intensity analyses. Treatment with 2-DG, DTT, TM, or BFA resulted in visibly swollen cells, while Mon- and TG-treated cells maintained a normal size similar to that of control *Chlorella* cells (Figure 4). These results indicate that treatment with 2-DG, DTT, TM, BFA, or Mon leads to TAG accumulation in UTEX 2714 cells, linking ER stress to lipid metabolism and storage in this microalga.

### 3.4. Time-Course Analysis Reveals Peak Lipid Accumulation at 48 h

To examine whether the effects of the selected chemical treatments were enhanced by prolonged exposure, we treated *C. sorokiniana* UTEX 2714 with 5 mM 2-DG, 5 μg/mL TM, 5 μM BFA, or 50 μM Mon for up to 120 h and measured Nile red fluorescence intensity as an indicator of lipid accumulation (Figure 5). The fluorescence intensity of cells treated with 2-DG, TM, and Mon peaked at 48 h and declined thereafter, suggesting a time-dependent reduction in TAG accumulation (Figure 5A). In contrast, BFA-treated cells showed a gradual increase in Nile red fluorescence over time. This trend may be attributed to a significant decrease in cell density (OD_750_) caused by BFA (Figure 5B), resulting in an apparent increase in OD-normalized fluorescence values. Based on these observations, 48 h post-treatment appears to be the most appropriate time point for assessing TAG accumulation in response to these chemical treatments for further study.

### 3.5. C. sorokiniana Strongly Accumulates TAG When Treated with 2-DG, BFA, TM, or Mon

To determine whether compounds reported to induce ER stress trigger TAG accumulation in *C. sorokiniana* UTEX 2714, we quantified lipid content using gas chromatography with flame-ionization detection (GC-FID). Based on Nile red staining and microscopy analysis, TG did not induce lipid droplet formation; therefore, only 1 μM TG was included in this assay. A previous study showed that BFA enhances TAG accumulation in *Chlorella* [21]; accordingly, we only tested 5 μM BFA. Likewise, since DTT has been reported to transiently increase lipid content after short-term treatment (4 h) in *Chlamydomonas* [17], we treated *C. sorokiniana* UTEX 2714 cells with 5 and 10 mM DTT under similar conditions for direct comparison. DMSO was used as the control condition for the TG, BFA, and Mon treatments. The results revealed a significant increase in both TAG content (mg/L) and TAG-to-total fatty acid ratio (TAG/tFA), a measure of the proportion of fatty acids diverted into TAG synthesis, in cells treated with 2-DG, DTT, TM, BFA, or Mon (Figure 6A,B). Notably, 10 mM 2-DG led to particularly strong responses, with TAG content and the TAG/tFA ratio increasing by 8.5-fold and 14.8-fold, respectively, compared to untreated control cells. Interestingly, TM and Mon, which reportedly induce no TAG accumulation in *Chlamydomonas* [17], caused a 6.2-fold and 8.9-fold increase, respectively, in the TAG/tFA ratio in *C. sorokiniana* cells. These findings suggest that compounds reported to induce ER stress may promote redistribution of fatty acids into storage lipids, reflecting a stress adaptation mechanism in *C. sorokiniana*.

### 3.6. Fatty Acid Composition Undergoes Dynamic Changes in Response to Treatment with Compounds Reported to Induce ER Stress

The UPR, triggered by ER stress, plays a critical role in regulating membrane lipid and fatty acid metabolism [14]. To investigate how compounds reported to induce ER stress affect fatty acid metabolism in *C. sorokiniana* UTEX 2714, we analyzed the fatty acid composition in tFA fractions following treatment with various compounds (Figure 7A). Treatment with 2-DG, DTT, TM, BFA, or Mon tended to increase the proportion of saturated fatty acids, such as 16:0, while reducing the levels of polyunsaturated fatty acids (PUFAs), including 16:3, 18:2, and 18:3. These compositional changes suggest that ER stress may induce fatty acid remodeling, characterized by enhanced saturation and PUFA depletion. In contrast, TG treatment caused only minor but statistically significant changes in certain fatty acids, showing a trend similar to DMSO treatment. This suggests that the observed metabolic shifts in response to TG or DMSO treatment are likely independent of strong ER stress responses and have minimal effects on overall cellular physiology.

We next compared the fatty acid composition of TAGs under these conditions. Treatment with 2-DG, DTT, TM, BFA, or Mon, which strongly induced TAG accumulation, resulted in TAG fatty acid profiles that were clearly distinct from those of tFAs. Specifically, while tFAs showed an increase in saturated fatty acids and a decrease in PUFAs, TAGs exhibited the opposite trend, with reduced or unchanged levels of saturated fatty acids and a marked increase in PUFAs (Figure 7B). In contrast, both the TAG and tFA profiles of TG-treated cells resembled those observed with DMSO treatment, suggesting that TG itself had no significant effect on lipid accumulation or fatty acid remodeling. Taken together, these findings highlight a strong correlation between the extent of TAG accumulation and the degree of fatty acid compositional change. The enrichment of PUFAs in TAGs suggests that these treatments promote sequestration of PUFAs into storage lipids. Overall, these results underscore the dynamic nature of lipid remodeling in *C. sorokiniana* and its association with the compounds reported to induce ER stress.

## 4. Discussion

### 4.1. Lipid Accumulation in C. sorokiniana Under ER Stress

Microalgae are promising bioresources because of their high productivity, favorable nutrient profiles, and sustainability. Among the many species used in various industries and research fields, *Chlorella* is particularly favored for its ease of cultivation and versatility. However, one major reason why *Chlorella* and other algae have not yet become practical major bioresources or alternative foods is their high production cost [22,23]. Therefore, elucidating the molecular mechanisms driving lipid accumulation is essential. In this study, we focused on promoting lipid accumulation through ER stress, a mechanism known to induce lipid storage [21]. Since ER stress is a downstream response to many environmental stressors, understanding this response could also provide insights into improving the stress tolerance of algae. In this study, using *C. sorokiniana* as a model, we first examined the response to six compounds reported to induce ER stress and found that *C. sorokiniana* cells exhibit growth sensitivity to 2-DG, DTT, TM, BFA, and Mon (Figure 2). Furthermore, treatments with these compounds were shown to induce significant TAG accumulation (Figure 6), likely as a mechanism to sequester PUFAs into lipid droplets (Figure 7). Notably, to our knowledge, this is the first report demonstrating that 2-DG can induce TAG accumulation in *C. sorokiniana*, highlighting a novel link between glucose analog treatment and lipid storage in microalgae. These results demonstrate a potential strategy for enhancing PUFA-enriched oil production in *C. sorokiniana* through modulation of ER stress pathways.

### 4.2. Effect of 2-DG on Lipid Droplet Formation in C. sorokiniana UTEX 2714

In this study, among the six tested compounds, 2-DG elicited significant accumulation of TAG within 48 h of treatment (Figure 6), resulting in an 8.5-fold increase in TAG content and a 14.8-fold increase in the TAG/tFA ratio. 2-DG, a glucose analog, is transported into cells via glucose transporters and phosphorylated by hexokinase to form 2-DG-6-phosphate. However, its inability to isomerize into fructose-6-phosphate leads to intracellular accumulation of 2-DG-6-phosphate, which disrupts N-linked glycosylation and induces ER stress in animal cells (Figure 1) [24,25]. In *C. sorokiniana*, 2-DG likely induces growth inhibition by promoting accumulation of unfolded proteins in the ER lumen, triggering the UPR. Interestingly, previous studies of *Chlamydomonas* reported no significant effect of 2-DG on growth or lipid content [17]. This divergence may be explained by the absence of glucose transporters in *Chlamydomonas* [9], in contrast to *C. sorokiniana* and other microalgae that can utilize glucose as a carbon source [26]. Our study revealed that *C. sorokiniana* exhibited chlorosis and significant growth defects when exposed to concentrations exceeding 1 mM 2-DG (Figure 2). Spotting assays and optical density measurements confirmed growth inhibition under these conditions (Figure 2). Despite these defects, cells treated with 2-DG accumulated a substantial amount of lipid droplets, as evidenced by microscopy and lipid staining (Figure 4 and Figure 6). These findings suggest that *C. sorokiniana* responds to 2-DG treatment, leading to increased TAG accumulation (Figure 6). Several reports have highlighted the broader metabolic impacts of 2-DG. For instance, 2-DG induces oxidative stress and lipid peroxidation, which disrupt lipid homeostasis in animals [27]. However, its role in promoting TAG accumulation has not been previously reported in *Chlorella*. Our lipid profiling results provide direct evidence of this novel effect, demonstrating the superior impact of 2-DG on TAG accumulation compared to other chemical treatments.

Given the pronounced TAG accumulation observed upon 2-DG treatment, we next sought to determine how this effect compares to other well-documented TAG-inducing conditions in *C. sorokiniana*. To contextualize these findings, we compared 2-DG–induced TAG accumulation with other known TAG-inducing conditions in *Chlorella*. For example, cells grown in high-salinity conditions (30 g/L NaCl) showed a 5.7-fold increase in TAG accumulation based on Nile red fluorescence intensity [28]. Similarly, *C. vulgaris* treated with the phytohormone ABA exhibited a 3.01-fold increase in lipid content [29]. In contrast, our GC-FID analysis demonstrated an 8.5-fold increase in TAG content (mg/L) in *C. sorokiniana* following 10 mM 2-DG treatment (Figure 6). These results highlight the remarkable efficacy of 2-DG in inducing TAG accumulation, surpassing that of other stressors or chemical treatments reported to date. Thus, our findings revealed that 2-DG is a potent inducer of TAG accumulation in *C. sorokiniana* through its induction of ER stress and subsequent lipid metabolism modulation. These insights offer new perspectives for leveraging 2-DG as a tool to enhance lipid production in microalgae for biotechnological applications.

### 4.3. Growth Inhibition and Lipid Accumulation Induced by BFA, Mon, DTT, and TM in C. sorokiniana UTEX 2714

In this study, *C. sorokiniana* UTEX 2714 exhibited significant growth defects and enhanced lipid accumulation when treated with the putative ER stress–inducing compounds BFA, Mon, DTT, and TM, along with 2-DG (Figure 6). DTT, TM, and 2-DG disrupted protein folding, while BFA and Mon affected vesicle trafficking. These disruptions triggered lipid metabolism changes, resulting in elevated TAG accumulation, likely as a stress adaptation mechanism. This highlights potential strategies for modulating lipid production in microalgae under stress conditions.

#### 4.3.1. Brefeldin A

BFA is a fungal metabolite known to interfere with vesicle trafficking between the ER and the Golgi apparatus. It inhibits anterograde membrane trafficking while promoting retrograde transport from the Golgi back to the ER, leading to the accumulation of proteins in the ER lumen [30,31]. This accumulation triggers ER stress, which subsequently induces lipid droplet formation as part of the cellular stress response. A previous report demonstrated that BFA treatment of *C. vulgaris* induces lipid droplet formation [17]. However, the study utilized a high concentration of BFA (75 µg/mL) with a short exposure time of 4 h and relied on Nile red staining for analysis [21]. In our study, BFA treatment in *C. sorokiniana* significantly reduced growth and viability while increasing TAG content 5.7-fold compared to the control (Figure 6). The accumulation of lipid droplets observed upon BFA treatment further supports its function as an inhibitor of vesicle trafficking and underscores its value as a tool to investigate the link between cellular stress responses and lipid metabolism. These findings underscore the critical role of vesicle trafficking in lipid biosynthesis and storage, establishing *C. sorokiniana* as a valuable model for exploring lipid remodeling under ER stress.

#### 4.3.2. Monensin

Mon is an ionophore that disrupts extracellular vesicle trafficking by interfering with ion gradients across membranes, thereby causing significant cellular stress. This disruption affects critical intracellular processes, such as protein sorting, secretion, and vesicle trafficking, ultimately triggering ER stress [32,33]. In *Chlamydomonas*, Mon treatment has been reported to induce cell aggregation, a characteristic stress phenotype, and upregulate the expression of ER stress marker genes, including those associated with the UPR [17]. These findings highlight the ability of Mon to modulate stress pathways and disrupt cellular homeostasis in microalgae. In the present study, Mon treatment caused chlorosis and significantly reduced cell viability in *C. sorokiniana*, accompanied by a notable increase in TAG accumulation, with a 5.5-fold increase in TAG content and an 8.9-fold increase in the TAG/tFA ratio (Figure 6). Interestingly, *C. sorokiniana* exhibited a much more severe response to Mon treatment compared to *Chlamydomonas*. At the same Mon concentration, *Chlamydomonas* cultures maintained a healthy green color and displayed only mild cell aggregation, without significant growth inhibition or TAG accumulation [17]. In contrast, *C. sorokiniana* showed pronounced chlorosis, reduced viability, and substantial TAG accumulation (Figure 2), indicating considerable interspecies variation in Mon sensitivity. This difference in the response between *Chlamydomonas* and *Chlorella* may result from differences in the sensitivity of the target protein of Mon or variations in the uptake mechanisms between these microalgae. Understanding the mechanisms by which vesicle transport inhibitors like BFA and Mon trigger lipid accumulation without directly inducing unfolded protein accumulation remains a key challenge in advancing our knowledge of lipid storage processes in microalgae.

#### 4.3.3. Tunicamycin

TM is a well-established inducer of ER stress, primarily disrupting N-linked glycosylation, a critical post-translational modification that occurs in the ER. This disruption leads to the accumulation of unfolded proteins in the ER lumen, triggering the UPR. In *Chlamydomonas*, our previous research demonstrated that TM-induced ER stress is mediated by the IRE1/bZIP1 signaling pathway, which plays a crucial role in regulating the biosynthesis of pinolenic acid (18:3Δ5,9,12), a PUFA associated with stress adaptation mechanisms [16]. In *C. sorokiniana* UTEX 2714, TM treatment caused significant growth defects, reduced viability, and increased the TAG content, revealing its effectiveness as an ER stress inducer. Notably, TM treatment also elevated the 18:3 content in UTEX 2714 cells (Figure 7), consistent with findings in *Chlamydomonas* [16]. Lipid analysis revealed that 5 µg/mL TM treatment induced a 6.5-fold increase in TAG content in UTEX 2714 cells (Figure 6), whereas *Chlamydomonas* exhibited no TAG accumulation under the same conditions [16,17]. This difference suggests that *Chlorella* may possess a distinct mechanism for channeling fatty acids into TAGs under ER stress, making it more efficient in lipid remodeling compared to *Chlamydomonas*.

#### 4.3.4. Dithiothreitol (DTT)

DTT is a potent reducing agent that induces ER stress by disrupting disulfide bond formation, a process essential for proper protein folding in the ER. This disruption leads to the accumulation of misfolded proteins and activation of the UPR. Unlike TM, which inhibits glycosylation, DTT primarily alters the oxidative environment of the ER, causing significant changes in subcellular redox status. In *C. sorokiniana*, DTT significantly increased the TAG/tFA ratio by 7.9-fold (Figure 6), indicating that *C. sorokiniana* cells actively prioritize TAG production under this treatment. DTT treatment also induced notable changes in tFA composition, characterized by an increase in saturated fatty acids, such as 16:0, and a decrease in PUFAs, including 18:3 and 16:3 (Figure 7). These changes suggest that the altered redox environment caused by DTT may have shifted cellular conditions toward a more oxidized state, influencing the fatty acid remodeling process. Indeed, DTT treatment induces oxidation stress in microalgae or plants [34]. These findings underscore the dependence of lipid remodeling responses on the nature of the ER stressor and its specific mode of action.

### 4.4. Fatty Acid Remodeling and TAG Accumulation Under ER Stress

According to our FA analysis, tFA and TAG composition displayed opposite trends in response to 2-DG, BFA, and Mon treatments. This contrast likely arises from the ER stress–induced preferential incorporation of PUFAs into TAGs. Under these conditions, *C. sorokiniana* cells appeared to accumulate TAGs esterified with PUFAs such as 18:2 and 18:3 (Figure 7). Previous research has demonstrated that *Chlorella* DGAT (diacylglycerol acyltransferase) enzymes exhibit substrate specificity toward PUFAs like 18:2 or 18:3 [35,36]. This specificity likely facilitates the sequestration of PUFAs localized at the ER membrane into lipid droplets. Under ER stress, this mechanism might play a crucial role in mitigating the risk of lipid peroxidation, a process in which reactive oxygen species (ROS) oxidize PUFAs, leading to cellular damage. By channeling PUFAs into lipid droplets, *Chlorella* cells might effectively reduce their exposure to oxidative stress. Indeed, previous studies have shown that lipid droplets serve as protective reservoirs, protecting FAs from peroxidation [37]. This mechanism becomes especially important under ER stress, where ROS levels tend to increase. These observations suggest a coordinated response in which ER stress triggers both protective lipid remodeling and lipid droplet biogenesis. However, the molecular signaling pathways that regulate this process in *Chlorella* remain largely unknown.

In model systems such as yeast and plants, the UPR is typically mediated by ER membrane–associated sensors including IRE1 (inositol-requiring enzyme 1) and bZIP transcription factors (e.g., bZIP60 in *Arabidopsis* and CrbZIP1 in *Chlamydomonas*), and in metazoans by PERK and ATF6 [16,38,39]. Upon accumulation of unfolded proteins in the ER lumen, IRE1 is activated and splices the mRNA of specific bZIP transcription factors, which then translocate to the nucleus to regulate downstream gene expression. These transcriptional programs include upregulation of ER chaperones, protein-folding enzymes, and, relevant to our study, enzymes involved in lipid biosynthesis and membrane remodeling. As for the molecular mechanisms underlying ER stress–induced lipid accumulation, studies in *Chlamydomonas* have shown that ER stress activates the UPR, leading to transcriptional remodeling of lipid metabolism. For example, the transcription factor CrbZIP1 is activated under ER stress and induces genes involved in ER membrane lipid biosynthesis, contributing to ER membrane expansion and homeostasis [16]. Additionally, the expression of TAG biosynthetic enzymes such as *DGAT1* (plastid-localized) and *DGTT1* (ER-localized) is upregulated in response to BFA treatment [17], supporting the idea that TAG accumulation functions as a protective response to ER stress by sequestering excess fatty acids into lipid droplets. However, ER stress signaling pathways and their downstream targets, including genes involved in lipid metabolism, have not yet been characterized in *Chlorella*. Further investigation is needed to determine whether similar UPR-regulated mechanisms exist in this genus. To address this, our future studies will focus on comparative transcriptomic analyses to identify conserved and species-specific regulatory pathways involved in the unfolded protein response and lipid metabolism. Additionally, employing reverse genetic approaches is essential to investigate the functional roles of candidate genes, including key UPR regulators. It is also necessary to elucidate the molecular mechanisms by which these putative ER stress inducers are involved in triggering ER stress. These studies will help elucidate the molecular mechanisms that govern lipid remodeling under ER stress in *Chlorella* and other industrially relevant microalgal species.

### 4.5. Industrial Potential and Optimization of ER Stress–Induced Lipid Accumulation in Chlorella

We demonstrated that inducing ER stress in *Chlorella* using ER stress–inducing compounds significantly enhances lipid accumulation (Figure 6). While the cost of these compounds is relatively high, this approach offers industrial advantages over traditional nutrient starvation strategies, such as nitrogen limitation. Nitrogen starvation poses several industrial challenges, in that it requires complex operational steps, including media replacement and cell recovery, which increase energy consumption and material costs [40,41,42]. Additionally, under nitrogen starvation, cell division ceases, meaning that maintaining or increasing culture biomass requires larger culture volumes and more extensive processing, significantly elevating production costs [5]. Moreover, achieving maximum TAG accumulation under nitrogen deficiency requires cells to be in the mid-log phase, restricting flexibility in biomass expansion [43]. In contrast, compounds that induce ER stress simplify the lipid accumulation process. The direct addition of stress-inducing compounds eliminates the need for media exchange, reducing operational complexity and costs. Although ER stress can negatively impact cell viability, maintaining a high initial cell density can mitigate growth inhibition while sustaining lipid productivity. This approach may not require specific growth-phase synchronization, allowing for more flexible biomass management. Additionally, ER stress–induced lipid accumulation may involve distinct metabolic pathways, including membrane lipid recycling, which could enhance lipid biosynthesis efficiency. This mechanism may offer an alternative route to increasing lipid yields without severely impairing photosynthetic capacity, unlike nitrogen starvation. Further optimization of ER stress conditions could provide further benefits for industrial lipid production in microalgae.

By elucidating the mechanisms underlying ER stress–induced lipid accumulation, it becomes possible to develop alternative strategies to regulate this pathway more efficiently. Understanding the genetic and metabolic networks involved in ER stress responses allows for the identification of new targets for CRISPR-based mutagenesis to enhance lipid accumulation without relying on costly chemical inducers. Gene editing approaches could be used to modify key regulatory genes involved in ER stress signaling, enabling microalgae to accumulate lipids under more favorable and cost-effective conditions. Additionally, this knowledge opens the possibility of alternative low-cost treatments that can induce ER stress without requiring expensive synthetic compounds. Importantly, ER stress signaling is also involved in responses to a variety of environmental stresses, such as heat, cold, and salinity. Modulating these pathways may improve microalgal tolerance to upstream environmental stressors, making them more suitable for large-scale cultivation. Large-scale microalgal cultivation still faces challenges related to cost, scalability, and process optimization, including high energy input, nutrient supply, and efficient harvesting techniques [44]. These advancements are particularly valuable for large-scale cultivation, as they offer a sustainable and economically feasible approach to enhancing lipid productivity, reducing dependency on costly chemical treatments, and improving the overall efficiency of industrial microalgal bioprocessing.

Beyond lipid production, *Chlorella* is a valuable source of multiple bioactive compounds, including carotenoids, starch, and biomass, further enhancing its industrial potential. *Chlorella* produces high-value carotenoids such as lutein and β-carotene, which possess antioxidant properties and have applications in the food and pharmaceutical industries due to their health benefits [1]. Moreover, certain *Chlorella* strains contain up to 37% starch, making them a promising raw material for bioethanol production [45]. Additionally, *Chlorella* biomass is widely used in various industries, including health foods and bioenergy. Considering these diverse applications, integrating ER stress–induced lipid accumulation with co-production of high-value biochemicals could significantly enhance the economic sustainability of microalgal biotechnology. This approach aligns with biorefinery principles, where multiple valuable products are extracted from a single biomass source, maximizing commercial viability while promoting sustainable industrial applications [45]. Further research into optimizing ER stress conditions and scaling up this method for industrial cultivation could facilitate the development of a cost-effective and scalable microalgal bioresource platform.

### 4.6. Environmental and Economic Considerations for Microalgal Biodiesel Production

Biofuels derived from microalgae, especially species like *Chlorella*, are being evaluated as a potential alternative for enhancing environmental sustainability. Evaluations based on lifecycle assessment (LCA) have been conducted to compare the environmental footprints of microalgal biofuels with those of conventional fossil diesel, taking into account the entire process from cultivation through fuel production. For instance, the global warming potential over a 100 year period (GWP100) indicates that fossil-derived diesel emits 8.84 × 10^−2^ kg CO_2_eq, whereas microalgal biofuel emits 1.48 × 10^−1^ kg CO_2_eq [46]. While these findings highlight the environmental promise of algal biofuels, lifecycle cost (LCC) analysis reveal that high operating costs remain a major challenge. Key cost drivers include the large volumes of water discharged during cultivation and the significant energy required for harvesting steps such as centrifugation [47]. To improve cost-effectiveness and increase lipid accumulation, optimizing cultivation and processing methods is essential. Among various strategies, chemical treatments have been reported as an effective strategy for large-scale algal cultivation [48]. Building on this approach, our research explores the application of ER stress–inducing compounds to promote lipid accumulation within algal cells, aiming to reduce reliance on downstream concentration processes. Although the current expense of such compounds poses a barrier, further insights into their mode of action could provide the way for cost-effective alternatives. Approaches such as non-GMO mutagenesis and lower-cost processing methods may offer viable paths forward, potentially lowering production costs while minimizing the need for energy-intensive operations like centrifugation.

## 5. Conclusions

In this study, we examined the effects of six compounds reported to induce ER stress on *C. sorokiniana* UTEX 2714 and compared their impact on growth phenotypes and lipid profiles (Figure 8). Among these compounds, 2-DG, DTT, TM, BFA, and Mon treatment significantly inhibited cell growth. Notably, 2-DG, DTT, TM, BFA, and Mon strongly promoted TAG synthesis (Figure 6). Furthermore, treatment with 2-DG, DTT, TM, BFA, or Mon decreased the proportion of PUFAs in tFAs while enriching them in TAGs (Figure 7). These findings provide critical insights into the role of ER stress in modulating lipid metabolism in *C. sorokiniana* UTEX 2714 and highlight potential strategies for enhancing TAG production with enriched PUFA content for applications in the sustainable bioresources and functional food industries.

## Figures and Tables

**Figure 1 bioengineering-12-00452-f001:**
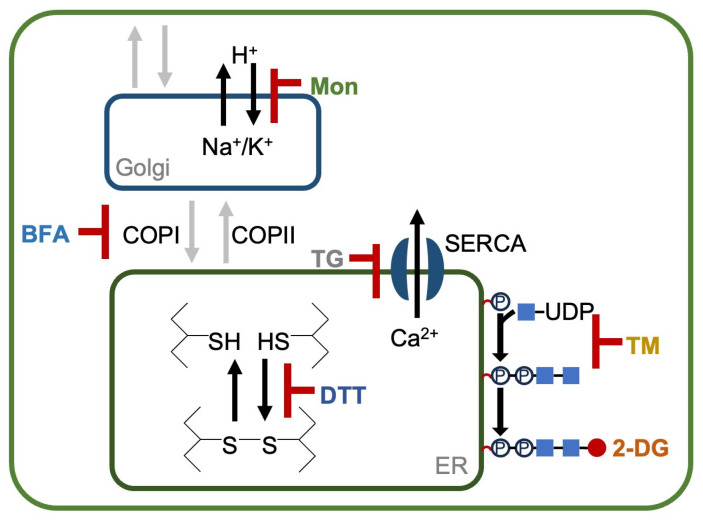
Functions of various ER stress–inducing compounds. 2-deoxyglucose (2-DG) is a synthetic glucose analog in which the C-2 hydroxyl group has been replaced with hydrogen. After entering the cell via glucose transporters, it inhibits glycolysis. Dithiothreitol (DTT) is an inhibitor of protein disulfide bond formation and can potentially activate the unfolded protein response (UPR). Tunicamycin (TM) induces ER stress by inhibiting N-linked glycosylation, leading to the accumulation of unfolded proteins in the ER. Thapsigargin (TG) disrupts ER Ca^2^⁺ homeostasis by inhibiting sarco/endoplasmic reticulum Ca^2^⁺-ATPase (SERCA). Brefeldin A (BFA) inhibits vesicle trafficking from the Golgi to the ER, causing accumulation of unfolded proteins in the ER. Lastly, monensin (Mon) is a Na⁺/H⁺ ionophore that disrupts the pH balance of both the ER and the Golgi.

**Figure 2 bioengineering-12-00452-f002:**
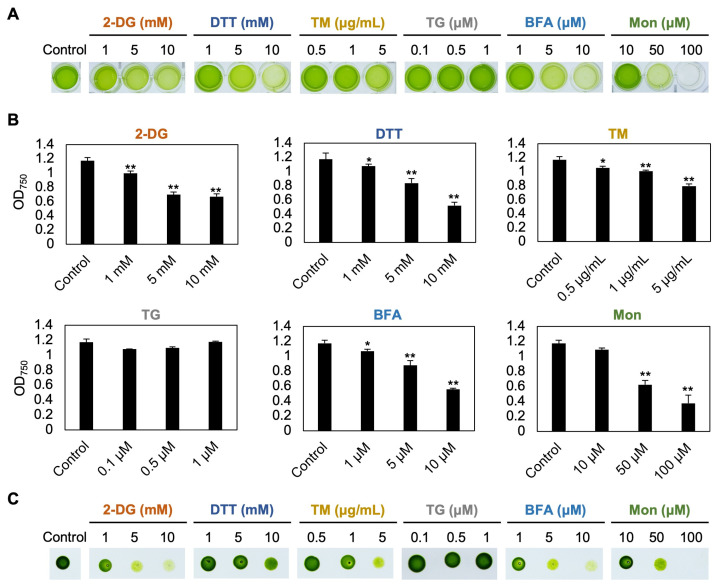
Growth phenotypes of *C. sorokiniana* UTEX 2714 treated with various concentrations of compounds reported to induce ER stress. (**A**) Photographs of *C. sorokiniana* UTEX 2714 cultures grown for 48 h in control TAP medium or TAP medium supplemented with the indicated concentrations of 2-DG, DTT, TM, TG, BFA, or Mon. (**B**) Optical density (OD_750_) of *C. sorokiniana* UTEX 2714 cultures after 48 h of growth under the same conditions. Error bars indicate the standard error (SE) of four biological replicates. Statistical significance was determined by *t*-test (* *p* < 0.1, ** *p* < 0.05). (**C**) Viability of *C. sorokiniana* UTEX 2714 cells after 48 h of treatment with the indicated compounds. Cells were spotted onto TAP medium to assess regrowth following treatment.

**Figure 3 bioengineering-12-00452-f003:**
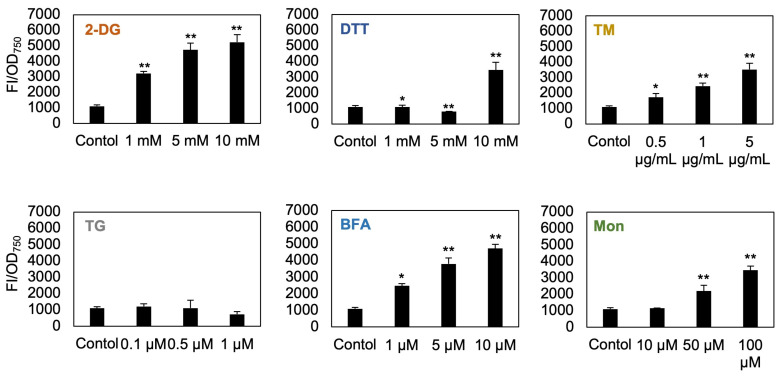
Lipid production in *C. sorokiniana* UTEX 2714 in response to compounds associated with ER stress. Nile red fluorescence in *C. sorokiniana* UTEX 2714 cultures grown for 48 h in control TAP medium or TAP medium containing the indicated concentrations of 2-DG, DTT, TM, TG, BFA, or Mon. Fluorescence intensity was normalized to OD_750_. Error bars indicate the standard error (SE) of three biological replicates. Statistical significance was determined by *t*-test (* *p* < 0.1, ** *p* < 0.05).

**Figure 4 bioengineering-12-00452-f004:**
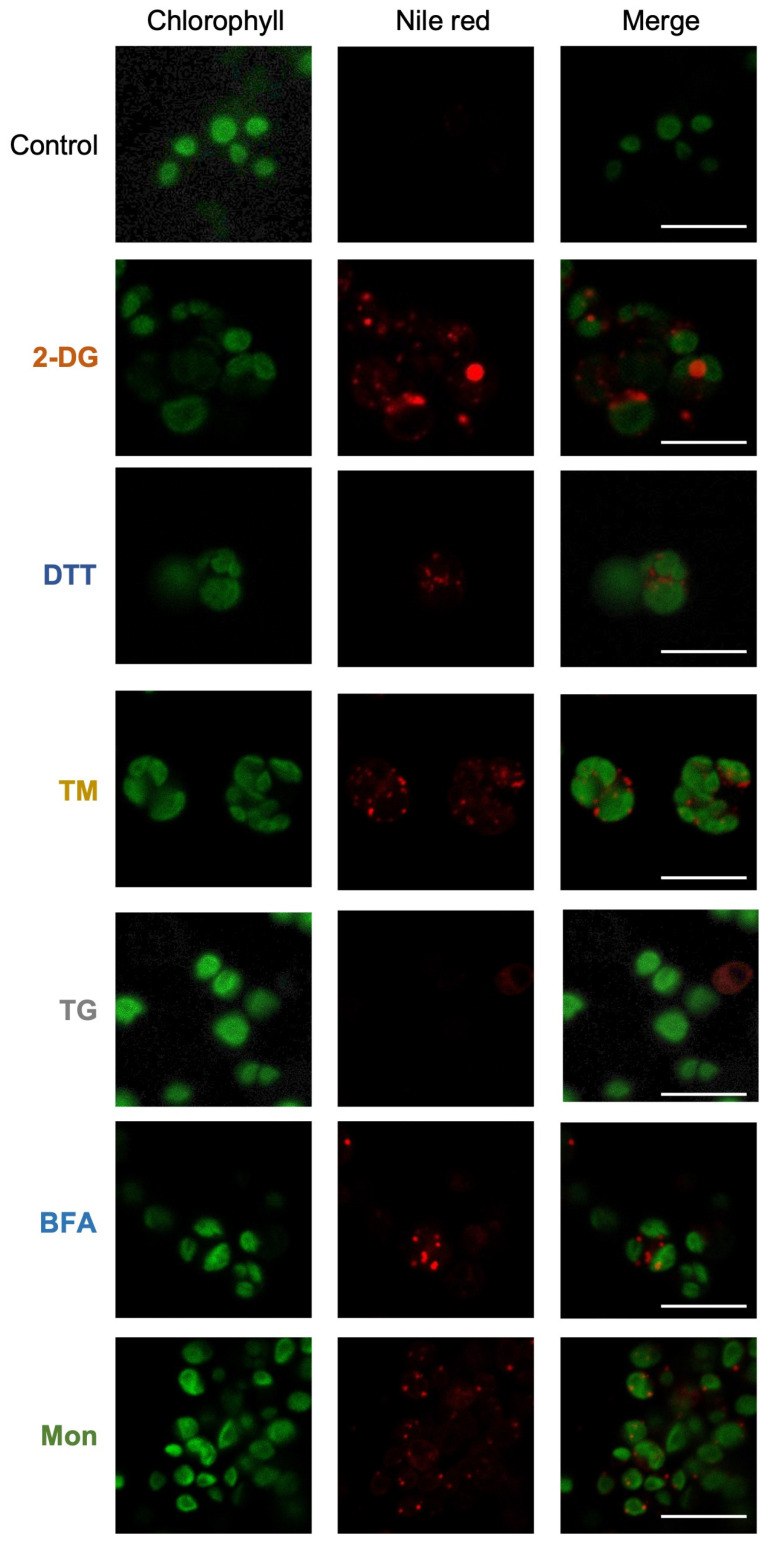
Lipid droplet formation in *C. sorokiniana* UTEX 2714 treated with ER stress–inducing compounds. Microscopy images of Nile red–stained *C. sorokiniana* UTEX 2714 cells grown in TAP medium with 5 mM 2-DG, 10 mM DTT, 5 μg/mL TM, 1 μM TG, 5 μM BFA, or 50 μM Mon for 48 h. Red represents Nile red fluorescence, marking lipid droplets, and green indicates chlorophyll autofluorescence. Scale bars, 10 μm.

**Figure 5 bioengineering-12-00452-f005:**
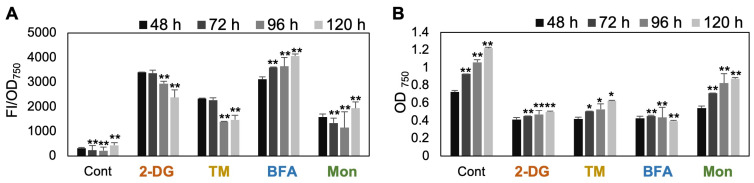
Lipid droplet accumulation peaks 48 h after treatment. (**A**) Nile red fluorescence intensity and (**B**) optical density (OD_750_) of *C. sorokiniana* UTEX 2714 cultures grown for up to 120 h in control TAP medium or TAP medium supplemented with 5 mM 2-DG, 5 μg/mL TM, 5 μM BFA, or 50 μM Mon. Nile red fluorescence intensity was normalized to OD_750_. Error bars indicate the standard error (SE) of three biological replicates. Statistical significance was determined by *t*-test (* *p* < 0.1, ** *p* < 0.05).

**Figure 6 bioengineering-12-00452-f006:**
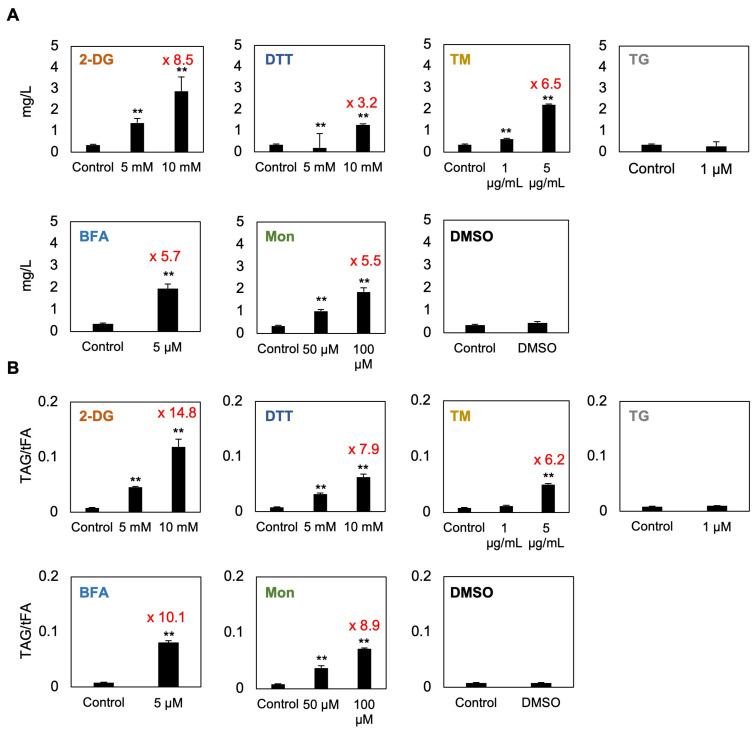
Triacylglycerol (TAG) accumulation in *C. sorokiniana* UTEX 2714 cells in response to compounds reported to induce ER stress. (**A**) TAG content in *C. sorokiniana* cells treated for 48 h with the indicated concentrations of 2-DG, TM, TG, BFA, Mon, or 10 µM DMSO or for 4 h with DTT. (**B**) TAG-to-total fatty acid (TAG/tFA) ratio in cells treated under the same conditions as in panel A. This ratio reflects the proportion of fatty acids redirected into storage lipids. Error bars indicate the standard error (SE) of four biological replicates from two independent experiments. Statistical significance was determined by *t*-test (** *p* < 0.05).

**Figure 7 bioengineering-12-00452-f007:**
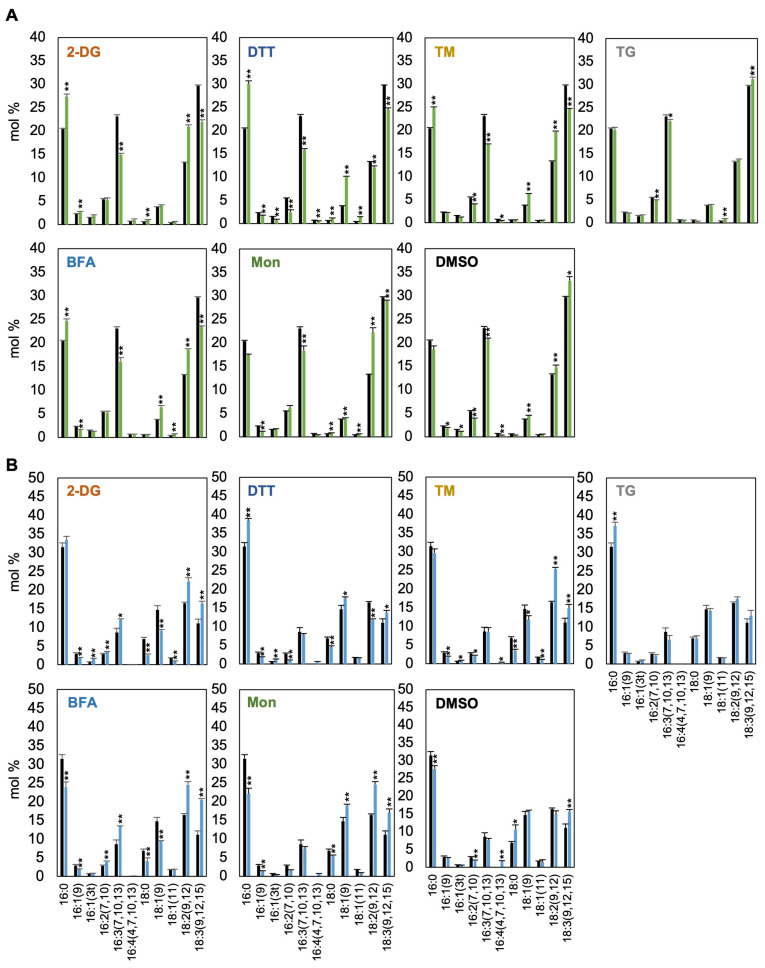
Compounds reported to induce ER stress alter total fatty acid (tFA) and triacylglycerol (TAG) composition in *C. sorokiniana*. (**A**) Fatty acid composition of tFAs in *C. sorokiniana* cells treated with 10 mM 2-DG, 5 μg/mL TM, 1 μM TG, 5 μM BFA, 50 μM Mon, or 10 µM DMSO for 48 h or 10 mM DTT for 4 h. Black, control; Green, treatment with putative ER stress inducers. (**B**) Fatty acid composition of triacylglycerols (TAGs) in cells treated under the same conditions. Black, control; Green, treatment with putative ER stress inducers. Error bars indicate the standard error (SE) from four biological replicates obtained from two independent experiments. Statistical significance was determined by *t*-test (* *p* < 0.1, ** *p* < 0.05).

**Figure 8 bioengineering-12-00452-f008:**
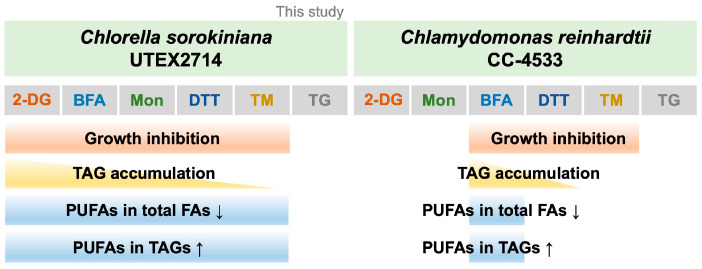
Proposed models of ER stress in *C. sorokiniana* UTEX 2714 and *Chlamydomonas*. (**Left**) In *C. sorokiniana*, 2-DG, BFA, Mon, DTT, and TM cause growth inhibition. In response to stress, 2-DG, BFA, Mon, DTT, and TM induce TAG accumulation while decreasing PUFAs in tFA. Conversely, PUFAs in TAG increase in response to 2-DG, BFA, Mon, DTT, and TM treatment. (**Right**) In *Chlamydomonas*, BFA, DTT, and TM cause growth inhibition. Only BFA and DTT induce TAG accumulation, with BFA altering PUFA composition in both tFA and TAG [17]. ↓ indicates a decrease, and ↑ indicates an increase.

## Data Availability

The raw data supporting the conclusions of this article will be made available by the authors on request.

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
