# Peer review of "Putative Endoplasmic Reticulum Stress Inducers Enhance Triacylglycerol Accumulation in Chlorella sorokiniana"

_bioengineering, 2025, doi:10.3390/bioengineering12050452_

Round 1

Reviewer 1 Report

Comments and Suggestions for Authors
Enhancing lipid synthesis of microlagae by simple chemical stimulators is of interesting for microlagae cultivation and products production. This manuscript investigated the influences of six ER  stress-inducing drugs on Chlorella grwoth, lipid accumulation and fatty acid profile.  The results demonstrated that different ER  stress-inducing drugs have different effect on growth and lipid composition.  This work is interesting and may provide  options for microalgae lipid synthesisi enhancement. 

Author Response

(Reviewer 1)

Comments and Suggestions for Authors

Enhancing lipid synthesis of microalgae by simple chemical stimulators is of interesting for microalgae cultivation and products production. This manuscript investigated the influences of six ER stress-inducing drugs on Chlorella growth, lipid accumulation and fatty acid profile. The results demonstrated that different ER stress-inducing drugs have different effect on growth and lipid composition.  This work is interesting and may provide options for microalgae lipid synthesis enhancement.

Response: Thank you for the positive feedback on the manuscript. The recognition that enhancing lipid synthesis in microalgae through simple chemical stimulators is an interesting approach is greatly appreciated. As noted, the study demonstrates that different ER stress-inducing drugs have distinct effects on Chlorella growth and lipid composition. It is encouraging that this work may provide potential options for enhancing microalgal lipid synthesis.

Reviewer 2 Report

Comments and Suggestions for Authors

Comment 1: The manuscript fails to provide a detailed understanding of the molecular mechanisms underlying ER stress-induced lipid accumulation in Chlorella sorokiniana. The authors should provide a more in-depth exploration of the signaling pathways and gene expression changes involved in this process.

Comment 2: The study lacks suitable control experiments to validate the specificity of the ER stress-inducing compounds. For instance, the authors should have included control treatments with non-specific inhibitors or solvent controls to rule out off-target effects.

Comment 3: The study only investigates the effects of ER stress on lipid metabolism in Chlorella sorokiniana, but does not provide a broader context or comparison with other microalgal species. The authors should expand their study to include a more comprehensive analysis of ER stress responses across different microalgal species.

Comment 4: The authors claim that Chlorella sorokiniana is one of the most widely cultivated species in industry, but fail to provide supporting evidence or citations. Please provide more context or references to substantiate this statement.

Comment 5: The authors state that nitrogen starvation is widely recognized as one of the most effective conditions to trigger TAG accumulation in microalgae, but neglect to discuss the potential limitations and drawbacks of this approach. Please provide a more balanced discussion of the advantages and disadvantages of nitrogen starvation.

Comment 6: The authors describe the role of the ER in protein folding, lipid biosynthesis, and maintaining cellular homeostasis, but fail to provide a clear explanation of how ER stress specifically impacts lipid metabolism. Please provide a more detailed explanation of the molecular mechanisms underlying ER stress-induced lipid accumulation.

Comment 7: The authors mention that the UPR pathway is activated under ER stress conditions, but do not provide a clear explanation of how this pathway regulates lipid metabolism. Please provide more insight into the specific roles of UPR components in lipid metabolism.

Comment 8: The authors describe the effects of 2-DG on cell growth, but fail to discuss the potential implications of this compound on cellular metabolism and energy production. Please provide a more comprehensive analysis of the effects of 2-DG on cellular metabolism.

Comment 9: The authors mention that ER stress influences not only protein homeostasis but also metabolic pathways, including lipid biosynthesis, but fail to provide a clear explanation of the specific molecular mechanisms involved. Please provide more insight into the molecular mechanisms underlying ER stress-induced lipid accumulation.

Comment 10: The authors mention the need for further investigation to determine whether the effects of ER stress are conserved across other microalgal species, but fail to provide a clear plan or strategy for addressing this knowledge gap. Please provide a more detailed description of the authors' plans for future research.

Comment 11: The authors mention the activation of the UPR pathway, but fail to provide a clear explanation of the specific components involved. Please provide more insight into the specific components of the UPR pathway and their roles in lipid metabolism.

Comment 12: The authors claim to have used Chlorella sorokiniana UTEX2714, but it is essential to provide the genetic background and history of the strain to ensure reproducibility.

Comment 13: The Tris-acetate-phosphate (TAP) medium composition and pH are not specified, which may affect the interpretation of the results.

Comment 14: The light intensity of 75 μmol photons m-2 s-1 is relatively low; have the authors considered using higher light intensities to simulate more realistic environmental conditions?

Comment 15: The constant shaking at 180 rpm may cause mechanical stress to the cells; have the authors controlled for this potential confounding factor?

Comment 16: The spot test is an unconventional method for assessing cell viability; the authors should consider using more established methods such as MTT or trypan blue assays.

Comment 17: The concentrations of ER stress drugs used in the spot test are not justified; have the authors performed a dose-response analysis to determine the optimal concentrations?

Comment 18: The incubation time of 48 hours is relatively short; have the authors considered longer incubation times to allow for more pronounced effects?

Comment 19: The Nile Red staining method is not quantitative; the authors should consider using more sensitive and specific methods such as GC-MS or TLC-FID to quantify lipid droplets. The fluorescence intensity (FI) measurement is not normalized to cell number or biomass; have the authors controlled for cell density and growth phase? The use of a microplate reader may not be suitable for measuring fluorescence intensity; have the authors considered using a more sensitive instrument such as a spectrofluorometer?

Comment 20: The lipid extraction method is not described in sufficient detail; the authors should provide a step-by-step protocol to ensure reproducibility.

Comment 21: The use of a TLC plate for lipid separation is outdated; have the authors considered using more modern and sensitive methods such as UPLC-MS or GC-MS?

Comment 22: The gas chromatograph conditions are not specified; the authors should provide the detailed instrument settings and conditions to ensure reproducibility.

Comment 23: The manuscript lacks a clear hypothesis and research question; the authors should reformulate their objectives to provide a clear direction for the study.

Comment 24: The study lacks a control group or baseline measurement; the authors should include a control group to provide a reference point for the ER stress drug treatments.

Comment 25: The manuscript does not provide any mechanistic insights into how ER stress induces lipid accumulation in Chlorella sorokiniana; the authors should include additional experiments to elucidate the underlying mechanisms.

Comments on the Quality of English Language

no

Author Response

Comment 1: The manuscript fails to provide a detailed understanding of the molecular mechanisms underlying ER stress-induced lipid accumulation in Chlorella sorokiniana. The authors should provide a more in-depth exploration of the signaling pathways and gene expression changes involved in this process.

Response: We thank the reviewer for this valuable comment. We fully agree that uncovering the molecular mechanisms, such as specific signaling pathways and gene expression profiles, underlying ER stress-induced lipid accumulation in Chlorella sorokinianais important for deepening our understanding of this process. Indeed, we initially attempted to include transcriptomic analysis to investigate these molecular responses. However, our preliminary data indicated that the stress signaling responses in Chlorella differ substantially from those reported in model organisms such as Chlamydomonas reinhardtii and Arabidopsis thaliana, suggesting the existence of species-specific regulatory mechanisms. Given these complexities, and the diversity of ER stress signaling across species, we concluded that a full molecular dissection lies beyond the current scope of this study. Instead, the current manuscript focuses on clearly demonstrating the reproducible phenotypic effects of the putative ER stress inducers, particularly 2-DG and tunicamycin, on lipid droplet formation in C. sorokiniana, which has not been well characterized before.

To address the reviewer’s concern, we have revised the term 'ER-stress inducer' to 'putative ER stress inducers' or 'compounds reported to induce ER stress' throughout the manuscript. We have also revised the manuscript to clarify that the detailed molecular mechanisms are not yet elucidated in Chlorella and will be addressed in future work. The following sentence has been added to the revised manuscript (Lines 527–):

" As for the molecular mechanisms underlying ER stress-induced lipid accumulation, studies in Chlamydomonas have shown that ER stress activates the UPR, leading to transcriptional remodeling of lipid metabolism. For example, the transcription factor CrbZIP1 is activated under ER stress and induces genes involved in ER membrane lipid biosynthesis, contributing to ER membrane expansion and homeostasis [16]. Additionally, the expression of TAG biosynthetic enzymes such as DGAT1 (plastid-localized) and DGTT1(ER-localized) is upregulated in response to BFA treatment [17], supporting the idea that TAG accumulation functions as a protective response to ER stress by sequestering excess fatty acids into lipid droplets. However, the ER stress signaling pathways and their downstream targets, including genes involved in lipid metabolism, have not yet been characterized in Chlorella. Further investigation is needed to determine whether similar UPR-regulated mechanisms exist in this genus. To address this, our future studies will focus on comparative transcriptomic analyses to identify conserved and species-specific regulatory pathways involved in the unfolded protein response and lipid metabolism. Additionally, we plan to employ reverse genetic approaches to investigate the functional roles of candidate genes, including key UPR regulators. It is also necessary to elucidate the molecular mechanisms by which these putative ER stress inducers are involved in triggering ER stress. These studies will help elucidate the molecular mechanisms that govern lipid remodeling under ER stress in Chlorella and other industrially relevant microalgal species."

Comment 2: The study lacks suitable control experiments to validate the specificity of the ER stress-inducing compounds. For instance, the authors should have included control treatments with non-specific inhibitors or solvent controls to rule out off-target effects.

Response: To address the concern, we have now included GC analysis using DMSO-treated samples as solvent controls, which are shown in the revised Figures 6 and 7. DMSO was used as the solvent for the ER stress-inducing compounds TG, BFA, and Mon in our experiments. We believe that this addition helps to validate the specificity of the observed phenotypes and strengthens the conclusions drawn from the treatment experiments.

Comment 3: The study only investigates the effects of ER stress on lipid metabolism in Chlorella sorokiniana, but does not provide a broader context or comparison with other microalgal species. The authors should expand their study to include a more comprehensive analysis of ER stress responses across different microalgal species.

Response: We appreciate the reviewer’s suggestion regarding the broader context of ER stress responses in microalgae. As noted, the current study focuses on Chlorella sorokiniana, a non-model but industrially relevant species. However, we would like to clarify that we have previously published a study on ER stress-induced lipid metabolism in the model alga Chlamydomonas reinhardtii (Je et al., 2023), which provides complementary insights into species-specific responses. In the present manuscript, we have incorporated a comparison of ER stress responses between Chlamydomonas and Chlorella, highlighting differences in stress sensitivity, and lipid droplet formation (Figure 8). We hope this clarifies that the current study builds upon previous work and contributes to a more comprehensive understanding of ER stress responses in diverse microalgal species.

Comment 4: The authors claim that Chlorella sorokiniana is one of the most widely cultivated species in industry, but fail to provide supporting evidence or citations. Please provide more context or references to substantiate this statement.

Response: We thank the reviewer for pointing out the lack of sufficient references to support our statement regarding the industrial relevance of Chlorella sorokiniana. Upon review, we acknowledge that the claim was not adequately supported by citations in the original manuscript. Therefore, we have removed the statement from the revised version to maintain accuracy and clarity.

Comment 5: The authors state that nitrogen starvation is widely recognized as one of the most effective conditions to trigger TAG accumulation in microalgae, but neglect to discuss the potential limitations and drawbacks of this approach. Please provide a more balanced discussion of the advantages and disadvantages of nitrogen starvation.

Response: According to the reviewer’s comments, we have now added a paragraph explaining the potential of ER stress-mediated lipid accumulation and how it compares to nutrient deficiency-induced lipid production. This additional discussion clarifies how ER stress may provide advantages despite potential growth rate restrictions, particularly in terms of lipid yield and metabolic efficiency.

4.5. Industrial potential and optimization of ER stress-induced lipid accumulation in Chlorella

We demonstrated that inducing ER stress in Chlorella using ER stress-inducing compounds significantly enhances lipid accumulation (Figure 6). While the cost of these compounds is relatively high, this approach offers industrial advantages over traditional nutrient starvation strategies, such as nitrogen limitation. Nitrogen starvation poses several industrial challenges, it requires complex operational steps, including media replacement and cell recovery, which increase energy consumption and material costs [40-42]. Additionally, under nitrogen starvation, cell division ceases, meaning that maintaining or increasing culture biomass requires larger culture volumes and more extensive processing, significantly elevating production costs [5]. Moreover, achieving maximum TAG accumulation under nitrogen deficiency requires cells to be in the mid-log phase, restricting flexibility in biomass expansion [43]. In contrast, compounds -induced ER stress simplifies the lipid accumulation process. The direct addition of stress-inducing compounds eliminates the need for media exchange, reducing operational complexity and costs. Although ER stress can negatively impact cell viability, maintaining a high initial cell density can mitigate growth inhibition while sustaining lipid productivity. This approach does not require specific growth-phase synchronization, allowing for more flexible biomass management. Additionally, ER stress-induced lipid accumulation may involve distinct metabolic pathways, including membrane lipid recycling, which could enhance lipid biosynthesis efficiency. This mechanism may offer an alternative route to increasing lipid yields without severely impairing photosynthetic capacity, unlike nitrogen starvation. Further optimization of ER stress conditions could provide further benefits for industrial lipid production in microalgae. (Lines 548-).”

Comment 6: The authors describe the role of the ER in protein folding, lipid biosynthesis, and maintaining cellular homeostasis, but fail to provide a clear explanation of how ER stress specifically impacts lipid metabolism. Please provide a more detailed explanation of the molecular mechanisms underlying ER stress-induced lipid accumulation.

Response: Thank you for your constructive comment. To address this point, we have revised the manuscript to include a more detailed explanation of the molecular mechanisms by which ER stress induces lipid accumulation. We have also added relevant references to support these mechanisms (Lines 527–).

“As for the molecular mechanisms underlying ER stress-induced lipid accumulation, studies in Chlamydomonas have shown that ER stress activates the UPR, leading to transcriptional remodeling of lipid metabolism. For example, the transcription factor CrbZIP1 is activated under ER stress and induces genes involved in ER membrane lipid biosynthesis, contributing to ER membrane expansion and homeostasis [16]. Additionally, the expression of TAG biosynthetic enzymes such as DGAT1 (plastid-localized) and DGTT1(ER-localized) is upregulated in response to BFA treatment [17], supporting the idea that TAG accumulation functions as a protective response to ER stress by sequestering excess fatty acids into lipid droplets. However, the ER stress signaling pathways and their downstream targets, including genes involved in lipid metabolism, have not yet been characterized in Chlorella. Further investigation is needed to determine whether similar UPR-regulated mechanisms exist in this genus. To address this, our future studies will focus on comparative transcriptomic analyses to identify conserved and species-specific regulatory pathways involved in the unfolded protein response and lipid metabolism. Additionally, we plan to employ reverse genetic approaches to investigate the functional roles of candidate genes, including key UPR regulators. It is also necessary to elucidate the molecular mechanisms by which these putative ER stress inducers are involved in triggering ER stress. These studies will help elucidate the molecular mechanisms that govern lipid remodeling under ER stress in Chlorella and other industrially relevant microalgal species”

Comment 7: The authors mention that the UPR pathway is activated under ER stress conditions, but do not provide a clear explanation of how this pathway regulates lipid metabolism. Please provide more insight into the specific roles of UPR components in lipid metabolism.

Response:

We thank the reviewer for this valuable comment. We agree that understanding how the UPR regulates lipid metabolism is crucial for elucidating the cellular response to ER stress.

In model organisms such as Chlamydomonas reinhardtii, ER stress activates the UPR, which includes transcriptional regulators like CrbZIP1. CrbZIP1 upregulates genes involved in ER membrane lipid biosynthesis, contributing to membrane expansion and cellular homeostasis (Yamaoka et al., 2019). Additionally, ER stress has been shown to induce the expression of TAG biosynthesis genes, including DGAT1 and DGTT1, facilitating the conversion of excess fatty acids into TAGs for storage in lipid droplets (Je et al., 2023). This response is thought to play a protective role by sequestering potentially toxic fatty acids and reducing lipid peroxidation. However, in Chlorella sorokiniana, the specific components of the UPR and their downstream targets remain uncharacterized. Although we attempted transcriptomic analysis, our preliminary data suggest that the UPR signaling pathway in Chlorella may differ significantly from those in Chlamydomonas or Arabidopsis, indicating the possibility of species-specific regulatory mechanisms. In response to the reviewer’s comment, we have revised the manuscript to clarify the known roles of UPR components in other systems and to explicitly state that the corresponding mechanisms in Chlorella remain to be elucidated. The following sentence has been added to the revised manuscript (Lines 535–):

“However, the ER stress signaling pathways and their downstream targets, including genes involved in lipid metabolism, have not yet been characterized in Chlorella. Further investigation is needed to determine whether similar UPR-regulated mechanisms exist in this genus. To address this, our future studies will focus on comparative transcriptomic analyses to identify conserved and species-specific regulatory pathways involved in the unfolded protein response and lipid metabolism. Additionally, we plan to employ reverse genetic approaches to investigate the functional roles of candidate genes, including key UPR regulators. It is also necessary to elucidate the molecular mechanisms by which these putative ER stress inducers are involved in triggering ER stress. These studies will help elucidate the molecular mechanisms that govern lipid remodeling under ER stress in Chlorella and other industrially relevant microalgal species.”

Comment 8: The authors describe the effects of 2-DG on cell growth, but fail to discuss the potential implications of this compound on cellular metabolism and energy production. Please provide a more comprehensive analysis of the effects of 2-DG on cellular metabolism.

Response: We agree that understanding the broader metabolic implications of 2-DG treatment is important. In response, we would like to clarify that we have already discussed the effects of 2-DG on cellular metabolism and energy-related processes in Section 4.2, titled "The effect of 2-DG on lipid droplet formation in C. sorokiniana UTEX2714." In this section, we explain that 2-DG, a glucose analog, enters the cell via glucose transporters and is phosphorylated by hexokinase to form 2-DG-6-phosphate. Due to its inability to isomerize into fructose-6-phosphate, 2-DG-6-phosphate accumulates in the cell and disrupts glycolytic flux and N-linked glycosylation, ultimately leading to ER stress. This disruption impacts cellular energy production and protein folding, which in turn affects cell growth and metabolic balance. Moreover, we discuss the specific impact of 2-DG on C. sorokiniana, including growth inhibition and chlorosis at concentrations above 1 mM, which are indicative of broader metabolic disturbance. We also reference prior studies that show 2-DG induces oxidative stress and lipid peroxidation, further demonstrating its influence on cellular metabolism. We believe this provides a sufficiently comprehensive analysis of how 2-DG affects not only lipid accumulation but also energy metabolism and cellular physiology in C. sorokiniana.

Comment 9: The authors mention that ER stress influences not only protein homeostasis but also metabolic pathways, including lipid biosynthesis, but fail to provide a clear explanation of the specific molecular mechanisms involved. Please provide more insight into the molecular mechanisms underlying ER stress-induced lipid accumulation.

Response: To address this point, we have revised the manuscript and added relevant references to clarify the molecular mechanisms through which ER stress regulates lipid metabolism (Lines 527–):

“As for the molecular mechanisms underlying ER stress-induced lipid accumulation, studies in Chlamydomonas have shown that ER stress activates the UPR, leading to transcriptional remodeling of lipid metabolism. For example, the transcription factor CrbZIP1 is activated under ER stress and induces genes involved in ER membrane lipid biosynthesis, contributing to ER membrane expansion and homeostasis [16]. Additionally, the expression of TAG biosynthetic enzymes such as DGAT1 (plastid-localized) and DGTT1(ER-localized) is upregulated in response to BFA treatment [17], supporting the idea that TAG accumulation functions as a protective response to ER stress by sequestering excess fatty acids into lipid droplets. However, the ER stress signaling pathways and their downstream targets, including genes involved in lipid metabolism, have not yet been characterized in Chlorella. Further investigation is needed to determine whether similar UPR-regulated mechanisms exist in this genus. To address this, our future studies will focus on comparative transcriptomic analyses to identify conserved and species-specific regulatory pathways involved in the unfolded protein response and lipid metabolism. Additionally, we plan to employ reverse genetic approaches to investigate the functional roles of candidate genes, including key UPR regulators. It is also necessary to elucidate the molecular mechanisms by which these putative ER stress inducers are involved in triggering ER stress. These studies will help elucidate the molecular mechanisms that govern lipid remodeling under ER stress in Chlorella and other industrially relevant microalgal species.”  

We hope this expanded explanation meets the reviewer’s expectations and enhances the clarity of the manuscript.

Comment 10: The authors mention the need for further investigation to determine whether the effects of ER stress are conserved across other microalgal species, but fail to provide a clear plan or strategy for addressing this knowledge gap. Please provide a more detailed description of the authors' plans for future research.

Response:

We appreciate the reviewer’s interest in the future directions of our research. While a detailed experimental plan falls beyond the scope of the current manuscript, we agree that it is important to investigate whether the mechanisms of ER stress-induced lipid accumulation are conserved across different microalgal species. To address this, our future studies will focus on comparative transcriptomic analyses to identify both conserved and species-specific regulatory pathways associated with the UPR and lipid metabolism. In parallel, we plan to employ reverse genetic approaches to functionally characterize candidate genes, such as DGAT isoforms and key UPR regulators. These investigations will provide further insight into the molecular basis of lipid remodeling under ER stress in Chlorella and other industrially relevant microalgal species. We have added a brief mention of this future research direction in the revised manuscript (Lines 538–):

“To address this, our future studies will focus on comparative transcriptomic analyses to identify conserved and species-specific regulatory pathways involved in the unfolded protein response and lipid metabolism. Additionally, we plan to employ reverse genetic approaches to investigate the functional roles of candidate genes, including key UPR regulators. It is also necessary to elucidate the molecular mechanisms by which these putative ER stress inducers are involved in triggering ER stress. These studies will help elucidate the molecular mechanisms that govern lipid remodeling under ER stress in Chlorella and other industrially relevant microalgal species.”

Comment 11: The authors mention the activation of the UPR pathway, but fail to provide a clear explanation of the specific components involved. Please provide more insight into the specific components of the UPR pathway and their roles in lipid metabolism.

Response: In response, we have revised the manuscript to provide a clearer and more detailed explanation of the specific components of the UPR pathway and their potential roles in lipid metabolism (Lines 519–):

“In model systems such as yeast and plants, the UPR is typically mediated by ER membrane-associated sensors including IRE1 (Inositol-Requiring Enzyme 1), bZIP transcription factors (e.g., bZIP60 in Arabidopsis and CrbZIP1 in Chlamydomonas), and in metazoans, PERK and ATF6 [16,38,39]. Upon accumulation of unfolded proteins in the ER lumen, IRE1 is activated and splices mRNA of specific bZIP transcription factors, which then translocate to the nucleus to regulate downstream gene expression. These transcriptional programs include upregulation of ER chaperones, protein-folding enzymes, and, relevant to our study, enzymes involved in lipid biosynthesis and membrane remodeling. As for the molecular mechanisms underlying ER stress-induced lipid accumulation, studies in Chlamydomonas have shown that ER stress activates the UPR, leading to transcriptional remodeling of lipid metabolism. For example, the transcription factor CrbZIP1 is activated under ER stress and induces genes involved in ER membrane lipid biosynthesis, contributing to ER membrane expansion and homeostasis [16]. Additionally, the expression of TAG biosynthetic enzymes such as DGAT1 (plastid-localized) and DGTT1 (ER-localized) is upregulated in response to BFA treatment [17], supporting the idea that TAG accumulation functions as a protective response to ER stress by sequestering excess fatty acids into lipid droplets. However, the ER stress signaling pathways and their downstream targets, including genes involved in lipid metabolism, have not yet been characterized in Chlorella. Further investigation is needed to determine whether similar UPR-regulated mechanisms exist in this genus. To address this, our future studies will focus on comparative transcriptomic analyses to identify conserved and species-specific regulatory pathways involved in the unfolded protein response and lipid metabolism. Additionally, we plan to employ reverse genetic approaches to investigate the functional roles of candidate genes, including key UPR regulators. It is also necessary to elucidate the molecular mechanisms by which these putative ER stress inducers are involved in triggering ER stress. These studies will help elucidate the molecular mechanisms that govern lipid remodeling under ER stress in Chlorella and other industrially relevant microalgal species.

Comment 12: The authors claim to have used Chlorella sorokiniana UTEX2714, but it is essential to provide the genetic background and history of the strain to ensure reproducibility.

Response: We agree that providing clear information about the strain background is important for ensuring reproducibility. In response to this comment, we have revised the manuscript to include a reference indicating the identification and classification of UTEX 2714 as Chlorella sorokiniana. Specifically, we have added the following sentence to the revised manuscript (Line 87-):

“In this study, we investigated the effects of six known ER stress-inducing compounds on the growth and lipid metabolism of Chlorella sorokiniana UTEX 2714, based on the hypothesis that ER stress promotes TAG accumulation through conserved stress response pathways. This strain was previously identified as Chlorella sorokiniana [19].”

Comment 13: The Tris-acetate-phosphate (TAP) medium composition and pH are not specified, which may affect the interpretation of the results.

Response: We agree that detailed information on the culture medium is important for reproducibility and proper interpretation of the experimental results. In response, we have now included the full composition of the Tris-acetate-phosphate (TAP) medium along with its initial pH in the Materials and Methods section of the revised manuscript.

Comment 14: The light intensity of 75 μmol photons m-2 s-1 is relatively low; have the authors considered using higher light intensities to simulate more realistic environmental conditions?

Response: In this study, we cultured Chlorella sorokiniana UTEX 2714 under continuous illumination at 75 μmol photons m-2s-1, which is a commonly used and well-established condition in studies involving Chlamydomonas reinhardtii. This light intensity allows for controlled growth and facilitates comparisons with previously published studies on ER stress and lipid metabolism. While we acknowledge that higher light intensities could better simulate certain environmental or industrial conditions, our study was designed to minimize variability and focus on the effects of ER stress-inducing compounds under a consistent baseline. Due to experimental and budgetary constraints, we were unable to explore multiple light intensities in this work. We fully agree that light intensity is an important factor influencing algal metabolism, and future studies should investigate how varying light conditions interact with ER stress responses and lipid accumulation.

Comment 15: The constant shaking at 180 rpm may cause mechanical stress to the cells; have the authors controlled for this potential confounding factor?

Response: In our experiments, we cultured Chlorella sorokiniana UTEX 2714 in small-volume flasks, glass tubes, or plastic plates under continuous shaking at 180 rpm. Based on preliminary trials, this condition was determined to be optimal for stable and homogeneous cultivation. When the shaking speed was either increased or decreased, we frequently observed cell aggregation or sedimentation, which adversely affected growth consistency and experimental reproducibility. While we acknowledge that constant shaking can potentially introduce mechanical stress in some systems, under our small-scale cultivation conditions, 180 rpm appears to strike an effective balance, minimizing such stress while ensuring adequate gas exchange and light distribution.

Comment 16: The spot test is an unconventional method for assessing cell viability; the authors should consider using more established methods such as MTT or trypan blue assays.

Response: While assays such as MTT and trypan blue staining are widely used for assessing cell viability in various systems, they often present limitations when applied to Chlorella species due to the presence of a rigid cell wall, which can hinder dye uptake and lead to ambiguous or misleading results. In contrast, the spot test we employed involves transferring treated cells onto fresh medium and monitoring their ability to grow under standard conditions. This method allows for the direct assessment of cell viability based on colony formation and regrowth.

Comment 17: The concentrations of ER stress drugs used in the spot test are not justified; have the authors performed a dose-response analysis to determine the optimal concentrations?

Response: The concentrations of ER stress-inducing compounds used in the spot test were initially selected based on previous studies in Chlamydomonas reinhardtii, which provided a useful reference for designing our experimental conditions. To further support the appropriateness of these concentrations, we have now performed lipid accumulation analyses for all tested conditions used in the spotting assay (Figures 6). These data clearly show that, for some compounds, lipid accumulation in Chlorella sorokiniana occurs in a dose-dependent manner, confirming the biological activity of the chosen concentrations and providing additional validation of our experimental setup.

Comment 18: The incubation time of 48 hours is relatively short; have the authors considered longer incubation times to allow for more pronounced effects?

Response: To evaluate whether a longer incubation period would enhance the observed effects, we conducted a time-course analysis of 2-DG, TM, BFA, and Mon treatment over a range of incubation times from 2 to 5 days. The results indicated that the extent of TAG accumulation was already evident at 48 hours and remained largely consistent at later time points, with no substantial increase in lipid accumulation beyond the 2-day mark. Based on these observations, we selected 48 hours as the representative incubation time, as it captures the primary effect of 2-DG while minimizing potential confounding factors such as nutrient depletion or secondary stress responses. We have revised the manuscript to briefly describe this rationale in the relevant section (Figure 5).

Comment 19: The Nile Red staining method is not quantitative; the authors should consider using more sensitive and specific methods such as GC-MS or TLC-FID to quantify lipid droplets. The fluorescence intensity (FI) measurement is not normalized to cell number or biomass; have the authors controlled for cell density and growth phase? The use of a microplate reader may not be suitable for measuring fluorescence intensity; have the authors considered using a more sensitive instrument such as a spectrofluorometer?

Response: We agree that Nile Red staining is not sufficient for precise lipid quantification and should be complemented by more sensitive and specific analytical methods. Thus, as already described in the original manuscript (Figure 6), we performed quantitative lipid analysis using gas chromatography with flame ionization detection (GC-FID). This method allowed us to accurately measure triacylglycerol (TAG) content and validate the results from the fluorescence-based screening. In this version, each 2 concentration was added to support the Nile red result.

Comment 20: The lipid extraction method is not described in sufficient detail; the authors should provide a step-by-step protocol to ensure reproducibility.

Response: We agree that Nile Red staining is not sufficient for precise lipid quantification and should be complemented by more sensitive and specific analytical methods. As already described in the original manuscript (Figure 6), we performed quantitative lipid analysis using gas chromatography with flame ionization detection (GC-FID). This method allowed us to accurately quantify triacylglycerol (TAG) content and validate the trends observed in the fluorescence-based screening. In the revised version of the manuscript, we have added data for two concentrations of each compound to further support the results obtained from the Nile red assay. This provides a clearer correlation between fluorescence intensity and TAG accumulation, reinforcing the validity of our initial screening approach.

Comment 21: The use of a TLC plate for lipid separation is outdated; have the authors considered using more modern and sensitive methods such as UPLC-MS or GC-MS?

Response: We appreciate the reviewer’s suggestion and fully acknowledge that techniques such as UPLC-MS and GC-MS offer high sensitivity and are valuable tools for lipid species identification and structural analysis. However, for the purpose of quantifying total fatty acid content and TAG accumulation, we selected GC-FID, which is widely regarded as one of the most accurate and reproducible methods for quantitative lipid analysis. Compared to MS-based techniques, FID provides superior linearity, stability, and quantification accuracy, especially when working with complex lipid mixtures or when absolute quantification is required (Jouhet et al., 2017). Therefore, while MS-based approaches are indeed powerful for lipidomics and molecular identification, we believe GC-FID is more suitable for the quantitative objectives of this study.

Ref: Jouhet J, Lupette J, Clerc O, Magneschi L, Bedhomme M, Collin S, Roy S, Maréchal E, Rébeillé F. LC-MS/MS versus TLC plus GC methods: Consistency of glycerolipid and fatty acid profiles in microalgae and higher plant cells and effect of a nitrogen starvation. PLoS One. 2017 Aug 3;12(8):e0182423.

Comment 22: The gas chromatograph conditions are not specified; the authors should provide the detailed instrument settings and conditions to ensure reproducibility.

Response: In response, we have now included the full GC-FID settings in the Materials and Methods section of the revised manuscript. The conditions are as follows (Lines 145-):

“A capillary column was temperature-programmed from 185°C to 230°C at 10°C min–1 with helium as carrier gas. One μL of sample was injected in a 270°C inlet with a 20:1 split ratio. Methyl ester derivatives were detected using an FID at 270°C”

Comment 23: The manuscript lacks a clear hypothesis and research question; the authors should reformulate their objectives to provide a clear direction for the study.

Response: Our primary objective was indeed to investigate how ER stress influences lipid metabolism in Chlorella sorokiniana UTEX2714, a species with industrial potential. Specifically, we aimed to determine whether ER stress-inducing compounds could promote triacylglycerol (TAG) accumulation and to explore potential mechanisms behind this response. However, we acknowledge that the research objective and hypothesis may not have been stated clearly enough in the original version of the manuscript. To improve clarity, we have revised the Introduction section to explicitly state our research question and the rationale behind testing six different ER stress-inducing compounds (Lines 87-):

" In this study, we investigated the effects of six known ER stress-inducing compounds on the growth and lipid metabolism of Chlorella sorokiniana UTEX 2714, based on the hypothesis that ER stress promotes TAG accumulation through conserved stress response pathways. This strain was previously identified as Chlorella sorokiniana [19]. "

Comment 24: The study lacks a control group or baseline measurement; the authors should include a control group to provide a reference point for the ER stress drug treatments.

Response: To address the concern, we have now included GC analysis using DMSO-treated samples as solvent controls, which are shown in the revised Figures 6 and 7. DMSO was used as the solvent for the ER stress-inducing compounds TG, BFA, and Mon in our experiments. We believe that this addition helps to validate the specificity of the observed phenotypes and strengthens the conclusions drawn from the treatment experiments.

Comment 25: The manuscript does not provide any mechanistic insights into how ER stress induces lipid accumulation in Chlorella sorokiniana; the authors should include additional experiments to elucidate the underlying mechanisms.

Response: To address the reviewer’s concern, we have revised the manuscript to clarify that the detailed molecular mechanisms are not yet elucidated in Chlorella and will be addressed in future work. The following sentence has been added to the revised manuscript (Lines 527–):

" As for the molecular mechanisms underlying ER stress-induced lipid accumulation, studies in Chlamydomonas have shown that ER stress activates the UPR, leading to transcriptional remodeling of lipid metabolism. For example, the transcription factor CrbZIP1 is activated under ER stress and induces genes involved in ER membrane lipid biosynthesis, contributing to ER membrane expansion and homeostasis [16]. Additionally, the expression of TAG biosynthetic enzymes such as DGAT1 (plastid-localized) and DGTT1(ER-localized) is upregulated in response to BFA treatment [17], supporting the idea that TAG accumulation functions as a protective response to ER stress by sequestering excess fatty acids into lipid droplets. However, the ER stress signaling pathways and their downstream targets, including genes involved in lipid metabolism, have not yet been characterized in Chlorella. Further investigation is needed to determine whether similar UPR-regulated mechanisms exist in this genus. To address this, our future studies will focus on comparative transcriptomic analyses to identify conserved and species-specific regulatory pathways involved in the unfolded protein response and lipid metabolism. Additionally, we plan to employ reverse genetic approaches to investigate the functional roles of candidate genes, including key UPR regulators. It is also necessary to elucidate the molecular mechanisms by which these putative ER stress inducers are involved in triggering ER stress. These studies will help elucidate the molecular mechanisms that govern lipid remodeling under ER stress in Chlorella and other industrially relevant microalgal species"

Reviewer 3 Report

Comments and Suggestions for Authors

Reviewer’s comments

Title: ER Stress-inducing drugs as a strategy for enhancing lipid accumulation in Chlorella sorokiniana

Manuscript Number: 3440770

Journal: Bioengineering

The research work entitled “ER Stress-inducing drugs as a strategy for enhancing lipid accumulation in Chlorella sorokiniana” compares six ER stress-inducing drugs for their ability to stimulate rapid-growing green algae Chlorella sorokiniana to produce high levels of triacylglycerol. This work has employed an in-depth investigation into the effects of varying doses of ER stress-inducing drugs on green algae physiology and changes in lipid production composition. However, there are several concepts and diagrams proposed by the author in the article that needs to be properly modified. I still believe that this work can provide a progressive contribution to the development of novel strategies and technologies for industrial mass production of algae oil. I suggested this work may be “minor revision” for publication in “Bioengineering”. Specific comments and general comments are given below:

Specific comments

1.      In the Introduction section, As the primary focus of this manuscript is enhancing the production of bio-based lipids in green algae, particularly triacylglycerol (TAG), it may be helpful to include 2-3 sentences in the introduction to emphasize the importance of lipid-based biomass and the critical role of TAG. This addition could provide readers with a clearer understanding of the broader significance of the study and its potential applications.

2.      It is suggested to use molar concentration as the unit for the six chemical agents to facilitate easier comparison of concentration differences between the drugs. If only TM is presented differently, it may not be ideal for consistency and clarity.

3.      For Fig. 5, it would be helpful to include the magnification and a scale bar in the enlarged area to improve readability.

4.      For Fig. 8, it might be helpful for the author to consistently set the maximum value of the Y axis to 50, as this would make it easier for readers to compare the data.

5.      In the Discussion section, the authors highlighted in the introduction that ER stress may be more efficient for lipid production compared to the previous nutrient deficiency-induced method, as nutrient deficiency can lead to growth delays and reduced photosynthetic efficiency. However, the article also suggests that green algae experience growth rate restrictions under ER stress. Therefore, it might be beneficial for the authors to clarify how microalgae may still have more advantages in producing lipids under ER stress compared to nutrient deficiency.

6.      In the Discussion section, based on current research, these ER stress-inducing drugs are relatively costly, and at this stage, they may not yet be a viable alternative for the industrial production of biofuels. However, the author presents a promising new research direction in this manuscript. It might be worth considering the potential of finding more affordable alternatives from natural extracts in the future. It is suggested that the author explore this concept further in the manuscript.

General comments

1.     Line 28-36, Please remove the instructions provided by the journal template.

2.     Line 168-169, Please remove extra spaces.

Author Response

Comments and Suggestions for Authors

The research work entitled “ER Stress-inducing drugs as a strategy for enhancing lipid accumulation in Chlorella sorokiniana” compares six ER stress-inducing drugs for their ability to stimulate rapid-growing green algae Chlorella sorokiniana to produce high levels of triacylglycerol. This work has employed an in-depth investigation into the effects of varying doses of ER stress-inducing drugs on green algae physiology and changes in lipid production composition. However, there are several concepts and diagrams proposed by the author in the article that needs to be properly modified. I still believe that this work can provide a progressive contribution to the development of novel strategies and technologies for industrial mass production of algae oil. I suggested this work may be “minor revision” for publication in “Bioengineering”. Specific comments and general comments are given below:

Specific comments

  1. In the Introduction section, As the primary focus of this manuscript is enhancing the production of bio-based lipids in green algae, particularly triacylglycerol (TAG), it may be helpful to include 2-3 sentences in the introduction to emphasize the importance of lipid-based biomass and the critical role of TAG. This addition could provide readers with a clearer understanding of the broader significance of the study and its potential applications.

Response: We have incorporated an explanation of the critical role of TAG in bio-based lipid production and its broader implications for the field as follows:

“Triacylglycerol (TAG) is a key microalgal lipid, serving as the primary energy storage molecule and an essential feedstock for biofuel production [2]. Microalgae can accumulate TAG under specific conditions, such as nitrogen starvation or high light intensity, making them promising candidates for sustainable biofuel applications [3]. Enhancing TAG productivity through metabolic engineering or optimized cultivation strategies could enable scalable and efficient biofuel production, reducing reliance on fossil fuels while supporting carbon-neutral energy solutions. (Lines 32-)”

  1. It is suggested to use molar concentration as the unit for the six chemical agents to facilitate easier comparison of concentration differences between the drugs. If only TM is presented differently, it may not be ideal for consistency and clarity.

Response: Tunicamycin a mixture of homologous nucleoside antibiotics rather than a single defined molecule. The use of µg/mL as a unit of measurement is essential, as it accounts for the total mass of all homologues in each volume, regardless of their individual molecular variations. This standardization is necessary to maintain consistent biological activity across different experimental setups.

  1. For Fig. 5, it would be helpful to include the magnification and a scale bar in the enlarged area to improve readability.

Response: We have replaced the previous fluorescence microscopy image with a clearer confocal image in Figure 6. This new image includes magnification information and improves readability, addressing the reviewer’s suggestion.

  1. For Fig. 8, it might be helpful for the author to consistently set the maximum value of the Y axis to 50, as this would make it easier for readers to compare the data.

Response: Thank you for your suggestion. We have adjusted the Y-axis scale in Figure 7, setting the maximum value consistently to 50% for FA composition in TAGs, as recommended.

  1. In the Discussion section, the authors highlighted in the introduction that ER stress may be more efficient for lipid production compared to the previous nutrient deficiency-induced method, as nutrient deficiency can lead to growth delays and reduced photosynthetic efficiency. However, the article also suggests that green algae experience growth rate restrictions under ER stress. Therefore, it might be beneficial for the authors to clarify how microalgae may still have more advantages in producing lipids under ER stress compared to nutrient deficiency.

Response: According to the reviewer’s comments, we have now added a paragraph explaining the potential of ER stress-mediated lipid accumulation and how it compares to nutrient deficiency-induced lipid production. This additional discussion clarifies how ER stress may provide advantages despite potential growth rate restrictions, particularly in terms of lipid yield and metabolic efficiency.

4.5. Industrial potential and optimization of ER stress-induced lipid accumulation in Chlorella

We demonstrated that inducing ER stress in Chlorella using ER stress-inducing compounds significantly enhances lipid accumulation (Figure 6). While the cost of these compounds is relatively high, this approach offers industrial advantages over traditional nutrient starvation strategies, such as nitrogen limitation. Nitrogen starvation poses several industrial challenges, it requires complex operational steps, including media replacement and cell recovery, which increase energy consumption and material costs [40-42]. Additionally, under nitrogen starvation, cell division ceases, meaning that maintaining or increasing culture biomass requires larger culture volumes and more extensive processing, significantly elevating production costs [5]. Moreover, achieving maximum TAG accumulation under nitrogen deficiency requires cells to be in the mid-log phase, restricting flexibility in biomass expansion [43]. In contrast, compounds -induced ER stress simplifies the lipid accumulation process. The direct addition of stress-inducing compounds eliminates the need for media exchange, reducing operational complexity and costs. Although ER stress can negatively impact cell viability, maintaining a high initial cell density can mitigate growth inhibition while sustaining lipid productivity. This approach does not require specific growth-phase synchronization, allowing for more flexible biomass management. Additionally, ER stress-induced lipid accumulation may involve distinct metabolic pathways, including membrane lipid recycling, which could enhance lipid biosynthesis efficiency. This mechanism may offer an alternative route to increasing lipid yields without severely impairing photosynthetic capacity, unlike nitrogen starvation. Further optimization of ER stress conditions could provide further benefits for industrial lipid production in microalgae. (Lines 548-).”

  1. In the Discussion section, based on current research, these ER stress-inducing drugs are relatively costly, and at this stage, they may not yet be a viable alternative for the industrial production of biofuels. However, the author presents a promising new research direction in this manuscript. It might be worth considering the potential of finding more affordable alternatives from natural extracts in the future. It is suggested that the author explore this concept further in the manuscript.

Response: According to the reviewer’s comments, we have now added the content related to natural extracts as more affordable alternatives and plan to include a section in our manuscript dedicated to investigating these possibilities. This will also involve reviewing relevant research to potentially identify cost-effective natural compounds.

“Beyond lipid production, Chlorella is a valuable source of multiple bioactive compounds, including carotenoids, starch, and biomass, further enhancing its industrial potential. Chlorella produces high-value carotenoids such as lutein and β-carotene, which possess antioxidant properties and have applications in the food and pharmaceutical industries due to their health benefits [1]. Moreover, certain Chlorella strains contain up to 37% starch, making them a promising raw material for bioethanol production [45]. Additionally, Chlorella biomass is widely used in various industries, including health foods and bioenergy. Considering these diverse applications, integrating ER stress-induced lipid accumulation with co-production of high-value biochemicals could significantly enhance the economic sustainability of microalgal biotechnology. This approach aligns with biorefinery principles, where multiple valuable products are extracted from a single biomass source, maximizing commercial viability while promoting sustainable industrial applications [45]. Further research into optimizing ER stress conditions and scaling up this method for industrial cultivation could facilitate the development of a cost-effective and scalable microalgal bioresource platform. (Lines 589-).”

General comments

  1. Line 28-36, Please remove the instructions provided by the journal template.

Response: We have removed the instructions provided by the journal template.

  1. Line 168-169, Please remove extra spaces.

Response: We have corrected the formatting by removing the extra spaces.

Reviewer 4 Report

Comments and Suggestions for Authors

The article focused on the stress-inducing compounds on Lipid production to increase biofuel yield. 

However, it needs some corrections. 

In line 30 and 31, the sentence does not make any sense. 

Improve the quality of Introduction section. Some addition work and previous papers are supposed to be added. Similar theme has already been covered by many author(s), so what is new in it?

Provide challenges as the substrate used by the author(s) since the sustainability and meeting large-scale production is still debatable. Add LCA study, economic feasibility, and sustainability in the given article. Materials and methods must be reasonable and it must contain only the relevant technique. 

In the current state, the article is a technical report. 

Author Response

Comments and Suggestions for Authors

The article focused on the stress-inducing compounds on Lipid production to increase biofuel yield. However, it needs some corrections.

  1. In line 30 and 31, the sentence does not make any sense.

Response: We have removed the instructions provided by the journal template as requested.

  1. Improve the quality of Introduction section. Some addition work and previous papers are supposed to be added. Similar theme has already been covered by many author(s), so what is new in it?

Response: We have added references to previous studies on ER stress-induced lipid accumulation in yeast and animals (Lines 74-).

“ER stress has been shown to initiate lipid droplet formation through several distinct, experimentally supported mechanisms across different organisms. In yeast, chemical stressors such as TM and BFA induce lipid droplet formation independently of the Ire1p pathway [12]. In mammalian systems, ER stress activates a transcriptional response involving SREBP-1c and other lipogenic regulators, thereby promoting TAG synthesis [13-15].”

Also, we have revised the Introduction section to better emphasize the novelty of our study. While previous studies reported lipid accumulation in Chlamydomonas under BFA and DTT treatment, and in Chlorella under various ER stress-inducing agents such as 2-DG, DTT, BFA, TM and Mon, our study uniquely demonstrates the efficient induction of lipid droplet formation by 2-DG, TM, and Mon in Chlorella. This highlights a distinct approach to promoting lipid production under ER stress conditions in this glucose-utilizing species.

“Treatments with 2-DG, DTT, TM, BFA, and Mon significantly inhibited cell growth. Lipid analysis revealed that 2-DG, DTT, TM, BFA, and Mon induced TAG accumulation in UTEX 2714 cells. Notably, we demonstrate for the first time that 2-DG efficiently induces TAG synthesis in Chlorella, a microalga capable of glucose uptake. These results revealed Chlorella-specific characteristics of lipid metabolism in response to putative ER stress compounds, thereby broadening our understanding of how ER stress influences lipid biosynthesis in microalgae. (Lines 90-).”

  1. Provide challenges as the substrate used by the author(s) since the sustainability and meeting large-scale production is still debatable.

Response: We now discuss a potential cost-effective strategy to improve the economic feasibility of ER stress-induced lipid accumulation in microalgae. By elucidating the mechanisms underlying this process, we can identify key genetic and metabolic regulators involved in ER stress responses. This understanding enables the development of alternative strategies to enhance lipid accumulation more efficiently, without relying on costly chemical inducers. Furthermore, since ER stress signaling intersects with pathways activated by various environmental stresses, this approach may also enhance stress tolerance, contributing to stable lipid production under fluctuating environmental conditions.

“By elucidating the mechanisms underlying ER stress-induced lipid accumulation, it becomes possible to develop alternative strategies to regulate this pathway more efficiently. Understanding the genetic and metabolic networks involved in ER stress responses allows for the identification of new targets for CRISPR-based mutagenesis to enhance lipid accumulation without relying on costly chemical inducers. Gene editing approaches could be used to modify key regulatory genes involved in ER stress signaling, enabling microalgae to accumulate lipids under more favorable and cost-effective conditions. Additionally, this knowledge opens the possibility of alternative low-cost treatments that can induce ER stress without requiring expensive synthetic compounds. Importantly, ER stress signaling is also involved in responses to a variety of environmental stresses, such as heat, cold, and salinity. Modulating these pathways may improve microalgal tolerance to upstream environmental stressors, making them more suitable for large-scale cultivation. Large-scale microalgal cultivation still faces challenges related to cost, scalability, and process optimization, including high energy input, nutrient supply, and efficient harvesting techniques [44]. These advancements are particularly valuable for large-scale cultivation, as they offer a sustainable and economically feasible approach to enhancing lipid productivity, reducing dependency on costly chemical treatments, and improving the overall efficiency of industrial microalgal bioprocessing (Lines 571-).”

  1. Add LCA study, economic feasibility, and sustainability in the given article.

Response: According to the reviewer’s comments, we have now added a paragraph explaining the LCA study, economic feasibility, and sustainability of using ER stress drugs to accumulate the lipid. This additional discussion clarifies how stress-inducing compounds on lipid production affect economic feasibility and sustainability.
4.6. Environmental and economic considerations for microalgal biodiesel production

Biofuels derived from microalgae, especially species like Chlorella, are being evaluated as a potential alternative for enhancing environmental sustainability. Evaluations based on Life-Cycle Assessment (LCA) have been conducted to compare the environmental footprints of microalgal biofuels with those of conventional fossil diesel, taking into account the entire process from cultivation through fuel production. For instance, the global warming potential over a 100-year period (GWP100) indicates that fossil-derived diesel emits 8.84 × 10−2 kg CO2eq, whereas microalgal biofuel emits 1.48 × 10−1 kg CO2eq [46]. While these findings highlight the environmental promise of algal biofuels, Life-Cycle Cost (LCC) analysis reveal that high operating costs remain a major challenge. Key cost drivers include the large volumes of water discharged during cultivation and the significant energy required for harvesting steps such as centrifugation [47]. To improve cost-effectiveness and increase lipid accumulation, optimizing cultivation and processing methods is essential. Among various strategies, chemical treatments have been reported as an effective strategy for large-scale algal cultivation [48]. Building on this approach, our research explores the application of ER stress-inducing compounds to promote lipid accumulation within algal cells, aiming to reduce reliance on downstream concentration processes. Although the current expense of such compounds poses a barrier, further insights into their mode of action could provide the way for cost-effective alternatives. Approaches such as non-GMO mutagenesis and lower-cost processing methods may offer viable paths forward, potentially lowering production costs while minimizing the need for energy-intensive operations like centrifugation. (Lines 605-)”.

  1. Materials and methods must be reasonable and it must contain only the relevant technique.

Response: We have carefully reviewed the materials and methods again and believe they are appropriate and relevant to our study. However, we would appreciate any specific points or concerns you may have regarding their reasonableness so that we can address them accordingly.

  1. In the current state, the article is a technical report.

Response: We respectfully disagree with the assessment that this manuscript constitutes a technical report. Technical reports are typically focused on documenting procedures, operational outcomes, or applied methods, and do not necessarily require novel findings or mechanistic insights. In contrast, this study presents new and biologically significant discoveries, fulfilling the criteria of a Research Article. Specifically, we demonstrate for the first time that ER stress-inducing compounds 2-DG and TM efficiently trigger lipid droplet formation in Chlorella. While previous studies have reported ER stress-induced lipid accumulation in Chlamydomonas using BFA and DTT, and in Chlorella under various agents, our work uniquely emphasizes the potent and reproducible effects of 2-DG and TM in Chlorella. In response to the reviewer’s comment, we have revised the Introduction to better clarify the novelty of our findings. Additionally, we explore the distinctive biological characteristics of Chlorella, particularly its ability to uptake glucose, which distinguishes it from Chlamydomonas and broadens its potential for metabolic engineering and biotechnological application. Taken together, the originality of the experimental findings, the mechanistic interpretation, and the relevance to applied algal biotechnology support that this manuscript is more appropriately classified as a Research Article. It not only contributes new knowledge to the field but also proposes a promising strategy for enhancing lipid production under ER stress, going well beyond the scope of a technical report.

Round 2

Reviewer 4 Report

Comments and Suggestions for Authors

Can be considered